



# Evaluation of the boundary layer dynamics of the TM5 model

E. N. Koffi[1], P. Bergamaschi[1], U. Karstens[2,3], M. Krol[4,5,6], A. Segers[7], M. Schmidt[8,11], I. Levin[8], A. T. Vermeulen[9,3], R. E. Fisher[10], V. Kazan[11], H. Klein Baltink[12], D. Lowry[10], G. Manca[1], H. A. J. Meijer[13], J. Moncrieff[14], S. Pal[15], M. Ramonet[11], H.A. Scheeren[1,13]

European Commission Joint Research Centre, Institute for Environment and Sustainability, Ispra (Va), Italy
Max-Planck-Institute for Biogeochemistry, Jena, Germany
ICOS Carbon Portal, ICOS ERIC at Lund University, Sweden
SRON Netherlands Institute for Space Research, Utrecht, Netherlands,
Institute for Marine and Atmospheric Research Utrecht, Utrecht, University, Utrecht, Netherlands
MAQ, Wageningen University and Research Centre, Wageningen, Netherlands,
Netherlands Organisation for Applied Scientific Research (TNO), Utrecht, Netherlands
Institut für Umweltphysik, Heidelberg University, Germany
Energy research Center Netherlands (ECN), Petten, Netherlands
Royal Holloway, University of London (RHUL), Egham, UK
Laboratoire des Sciences du Climat et de l'Environnement, LSCE/IPSL, CEA-CNRS-UVSQ, Université Paris-Saclay, F-91191 Gif-sur-Yvette, France
Royal Netherlands Meteorological Institute (KNMI), Netherlands
Centrum voor Isotopen Onderzoek (CIO), Rijksuniversiteit Groningen, Netherlands
Atmospheric Chemistry Research Group, University of Bristol, UK
Department of Meteorology, Pennsylvania State University, State College, PA, USA

01 March 2016

submitted to Geosci. Model Dev.



**Abstract**
We evaluate the capability of the global atmospheric transport model TM5 to reproduce
observations of the boundary layer dynamics and the associated variability of trace gases close to
the surface, using radon ($^{222}$Rn), which is an excellent tracer for vertical mixing owing to its
short lifetime (half-life) of 3.82 days. Focusing on the European scale, we compare the boundary
layer height (BLH) in the TM5 model with observations from the NOAA Integrated Global
Radiosonde Archive (IGRA) and in addition with ceilometer measurements at Cabauw (The
Netherlands) and lidar BLH retrievals at Trainou (France). Furthermore, we compare TM5
simulations of $^{222}$Rn activity concentrations, using a novel, process-based $^{222}$Rn flux map over
Europe (Karstens et al., 2015), with quasi-continuous $^{222}$Rn measurements from 10 European
monitoring stations.
The TM5 model reproduces relatively well the daytime BLH (within ~10-20% for most of the
stations), except for coastal sites, for which differences are usually larger due to model
representation errors. During night, TM5 overestimates the shallow nocturnal BLHs, especially
for the very low observed BLHs ($< 100$ m) during summer.
The $^{222}$Rn activity concentration simulations based on the new $^{222}$Rn flux map show significant
improvements especially regarding the average seasonal variability, compared to simulations
using constant $^{222}$Rn fluxes. Nevertheless, the (relative) differences between simulated and
observed daytime minimum $^{222}$Rn activity concentrations are larger for several stations (on the
order of 50%) compared to the (relative) differences between simulated and observed BLH at
noon. Although the nocturnal BLH is often higher in the model than observed, simulated $^{222}$Rn
nighttime maxima are larger at several continental stations, which points to potential deficiencies
of TM5 to correctly simulate the vertical gradients within the nocturnal boundary layer,
limitations of the $^{222}$Rn flux map, or issues related to the definition of the nocturnal BLH.
At several stations the simulated decrease of $^{222}$Rn activity concentrations in the morning is
faster than observed. In addition, simulated vertical $^{222}$Rn activity concentration gradients at
Cabauw decrease faster than observations during the morning transition period, and are in
general lower than observed gradients during daytime, which points to too fast vertical mixing in
the TM5 boundary layer during daytime. Furthermore, the capability of the TM5 model to
simulate the diurnal BLH cycle is limited due to the current coarse temporal resolution (3hr/6hr)
of the TM5 input meteorology.
Additionally, we analyze the impact of a new treatment of convection in TM5, based on the
ECMWF reanalysis, leading to overall significantly lower (on the order of ~20%) surface $^{222}$Rn
activity concentrations during daytime compared to the current default convection scheme based
on Tiedtke (1989). However, the performance of the model simulations compared to the $^{222}$Rn
observations is very similar in terms of root mean square and correlation coefficient for both
convection schemes.



# 1. Introduction

The boundary layer, the lowest portion of the atmosphere, is largely affected by the Earth's surface forcing. This layer is usually separated from the free troposphere (where the surface effects are weak) by a thin and strong stable layer (capping inversion) that traps turbulence, moisture, and trace gases in the boundary layer. The thickness of the boundary layer is variable in space and time and can range from tens of meters to 4 km, depending on both the synoptic and local meteorological conditions (Stull, 1988). The height of the boundary layer is an essential parameter in atmospheric transport models, since it controls the extent of the vertical mixing of trace gases emitted near the surface. The ability of global transport models to reproduce the boundary layer dynamics has been investigated earlier (e.g., Denning et al., 1999; Dentener et al., 1999). The authors have recommended the use of both high temporal resolution of meteorological data within the lower levels (Dentener et al., 1999) and fine horizontal and vertical resolutions (Krol et al., 2005) for a better reproduction of the meso-scale processes in the model. The realistic simulation of the boundary layer height (BLH) is crucial especially for inverse modelling simulations that aim at estimating surface fluxes from observed concentrations. This is the case in particular for regional flux inversions which make use of regional concentration measurements that capture the signal from regional sources (and sinks). Regional inversions of greenhouse gases (GHG) ($CO_2$, $CH_4$, $N_2O$, halocarbons) were reported especially for Europe and North America, making use of the increasing number of regional monitoring stations in these areas (e.g., Gerbig et al., 2003; Carouge et al., 2008; Kort et al., 2008; Bergamaschi et al., 2010; Corazza et al., 2011; Manning et al., 2011; Broquet et al., 2013; Bergamaschi et al., 2015; Ganesan et al., 2015).

In order to evaluate the quality of such flux inversions, a thorough validation of the applied atmospheric transport models is essential. In this study we present a detailed evaluation of the boundary layer dynamics of the TM5 model (Krol et al., 2005), which is the global transport model used in the TM5-4DVAR inverse modelling framework (Meirink et al., 2008), applied in several of the European inversions mentioned above (Corazza et al., 2011; Bergamaschi et al., 2010; 2015). In a first step, we compare the model BLH with the sounding-derived BLH of the NOAA Integrated Global Radiosonde Archive (IGRA) (Seidel et al., 2012) at European scale. Radiosonde data have been considered to give the most accurate BLHs (Collaud Coen et al., 2014). The model BLHs are also compared to those derived from the ceilometer and lidar measurements at two European stations (Cabauw and Trainou). As a second step, we compare TM5 simulations of $^{222}$Rn activity concentrations with measurements at 10 European stations. $^{222}$Rn is an excellent tracer for boundary layer mixing due to its short lifetime (half-life) of 3.82 days and has been widely used for model validation (e.g., Jacob and Prather, 1990; Jacob et al., 1997; Dentener et al., 1999; Chevillard et al., 2002; Taguchi et al., 2011). However, the use of $^{222}$Rn for this purpose has been limited by the simplified assumption of constant $^{222}$Rn fluxes over land used in most $^{222}$Rn validation studies published so far. It has also been limited by the fact that the observed $^{222}$Rn activity concentrations from different stations were not harmonized. Here, we make use of a novel detailed $^{222}$Rn flux map over Europe (Karstens et al., 2015) based on a parameterization of $^{222}$Rn production and transport in the soil as well as improved observed $^{222}$Rn activity concentrations obtained through a detailed comparison study (Schmithüsen et al., 2016). The development of this $^{222}$Rn flux map has been performed within the European project InGOS ('Integrated non-$CO_2$ Greenhouse gas Observing System'), including also a comparison



of different transport models (including TM5). While this model comparison will be published elsewhere (Karstens et al., 2016, manuscript in preparation), we present here the analysis for the TM5 model aiming at the identification and quantification of potential systematic errors in the simulation of the BLH dynamics, which could directly translate into systematic errors in the derived surface fluxes. Our study also includes the evaluation of a new parameterization of convection in TM5, based on ECMWF (re)analysis, compared to the default convection scheme used so far, based on the parameterization of Tiedtke (1989).

**2. Observations**

2.1. Boundary layer height

*2.1.1. IGRA data*

We use BLHs of the NOAA IGRA database, which covers the 1990-2010 period (Seidel et al., 2012). The IGRA data is based on radiosonde measurements that are usually released at 00 and 12 UTC. The IGRA radiosonde network over Europe is presented in **Figure 1**. The dynamic (wind speed and direction) and thermal (temperature and humidity) profiles from the radiosondes are utilized to compute BLHs using the bulk Richardson number method [Eq.1; Section 3.2]. In these BLH calculations both the surface wind (i.e., $u_s$ and $v_s$ in Eq.1) and the surface friction velocity ($u^*$) are unknown and set to zero. The critical value of the bulk Richardson number ($R_{ic}$) is set to 0.25 (instead of 0.3 as used in TM5; see Section 3.2). These settings for the IGRA database were also adopted in the InGOS protocol for the evaluation of the transport models involved in InGOS inverse modelling analyses (Karstens et al., 2016, manuscript in preparation). The methodological uncertainties in the IGRA BLH data were evaluated based on paired soundings released at the same site (Seidel et al., 2012). Results show that the choice of $R_{ic}$ does not introduce large uncertainty, but other methodological choices (including surface wind speed estimates and vertical interpolation of the bulk Richardson number profile) as well as the vertical resolution of the sounding data are larger sources of uncertainty in the derived BLHs (Seidel et al., 2012). The authors reported relative uncertainties in the IGRA BLHs that can be large (>50%) for shallow BLHs (< 1 km; mainly observed during night or early in the morning), but much smaller (usually <20%) for deep BLHs (> 1 km) during daytime.

*2.1.2. Lidar and ceilometer data*

The principle of LIDAR (LIght Detection And Ranging; hereafter lidar) is based on a pulsed laser light emitted into the atmosphere which is back-scattered by aerosol particles and molecules. The lidar algorithms derive the BLHs by searching the location of the strongest aerosol gradient in the vertical dimension (e.g., Haeffelin et al., 2011; Pal et al., 2012; Griffiths et al., 2013; Pal et al., 2015). A ceilometer is a 'low-cost lidar' which was initially used for the detection of cloud base heights. However, since the backscatter signal of aerosols is lower than that of clouds, the sensitivity of ceilometers in retrieving the boundary layer height is much less than that of lidar instruments (Pal, 2014). In contrast to IGRA data (i.e., radiosonde based BLH), the ceilometer and lidar allow measurements of the diurnal BLH cycle. However, the algorithms of both lidar and ceilometer have some difficulties to assign the BLH during night and tend to wrongly attribute the height of the residual layer of aerosol (often with larger signal) as the real





mixed layer (e.g., Angevine et al., 1998; Eresmaa et al., 2006; Haij et al., 2006). Lidar/ceilometer nocturnal BLHs are also higher due to the fact that their overlap height can be above the nocturnal shallow BLH (Pal et al., 2015). Uncertainties in lidar retrieved BLHs were assessed based on a comparison between radiosonde based BLHs and wavelet derived BLH estimates from lidar and found to be about 60 m (Pal et al., 2013).

We use the BLHs retrieved from lidar and ceilometer measurements at Trainou and Cabauw, respectively (see **Figure 1** for their locations). The lidar (ALS-300) measurements at Trainou are described by Pal et al. (2012). The ceilometer at Cabauw is part of the network of the Vaisala LD-40 ceilometer in the Netherlands operated by the Royal Netherlands Meteorological Institute (KNMI; Haij et al., 2006). We analyze the ceilometer measurements at Cabauw for 2010 and the lidar data at Trainou for 2011. For Cabauw we compare the ceilometer based BLH for 2010 with the BLH data from the closest IGRA station (De Bilt), with results at 12 UTC shown in **Figure 2**.

2.2. Observed $^{222}$Rn activity concentrations

The observed $^{222}$Rn activity concentrations are obtained from 2 different measurement methods:
 (1) The 'two-filter' method developed by the Australian Nuclear Science and Technology Organization (ANSTO) (Whittlestone and Zahorowski, 1998). After drawing the sampled air continuously through a delay volume to let all short-lived $^{220}$Rn in the sampled air decay, it passes through a first filter that removes all $^{222}$Rn and $^{220}$Rn decay products. Filtered air then enters a delay chamber in which new $^{222}$Rn progeny ($^{218}$Po and $^{214}$Po) are produced. A second flow loop within the delay chamber passes the air through a second filter, which collects the new $^{222}$Rn progeny formed under controlled conditions. Hence, in the ANSTO system $^{222}$Rn activity concentration is measured directly through its newly formed progeny in the sampled air (Whittlestone and Zahorowski, 1998; Zahorowski et al., 2004). In routine operation, ANSTO monitors are calibrated monthly by injecting $^{222}$Rn from a well characterized (to about ±4%) $^{226}$Radium source. For ambient air measurements at 1 Bq m$^{-3}$ activity concentration, the total uncertainty of hourly measurements is of order 10%, which includes uncertainty in flow rate as well as counting statistics.

(2) The one-filter methods used at the European stations are all based on the collection of the short-lived $^{222}$Rn and $^{220}$Rn ($^{212}$Pb) decay products, which are attached to aerosols. These decay products are accumulated on either static or moving aerosol filters and measured by α or β spectroscopy (see references given in **Table 1**). In order to derive the atmospheric $^{222}$Rn activity concentration, this method requires corrections for the atmospheric radioactive disequilibrium between the measured $^{222}$Rn daughters, $^{214}$Po and/or $^{218}$Po and $^{222}$Rn.

We use $^{222}$Rn activity concentration measurements from 10 European stations over the 2006-2011 period (**Figure 1** and **Table 1**). The data from the different stations have been harmonized based on an extensive comparison study performed within the InGOS project (Schmithuesen et al., 2016). Based on the tall tower measurements at Cabauw and Lutjewad conducted at different heights above ground level as well as on an earlier comparison at Schauinsland station (Xia et



al., 2010) and new comparison measurements in Heidelberg with an ANSTO system, correction factors for disequilibrium have also been estimated (Schmithuesen et al., 2016). All data used in the present study have been corrected accordingly and brought to a common ANSTO scale. A typical uncertainty of $^{222}$Rn data from the different one-filter systems, including the uncertainty of the disequilibrium is estimated to ±10 to 15%.

At the monitoring station Ispra, $^{222}$Rn activity concentration has been measured using an ANSTO instrument, sampling air at an inlet positioned at 3.5m above the ground, close to the GHG-sampling mast with a height of 15m. Recent additional $^{222}$Rn measurements using the 15m inlet of the GHG mast (employing an Alphaguard PQ2000 (Genitron) instrument, calibrated against the ANSTO monitor) revealed significant differences of the $^{222}$Rn activity at the two sampling heights during periods with low wind speeds. These differences showed that there are significant vertical $^{222}$Rn gradients close to the ground. Based on the comparison of the two sampling heights during a 3-month period, we derive a wind-speed dependent correction, in order to 'normalize' the entire time series of the ANSTO measurements (at 3.5m above ground) to the 15m inlet, which is considered to be more representative. The uncertainty of this wind-speed dependent correction (based on the 1σ standard deviation during the 3-month comparison) is included in the time series shown in the Supplement (**Figure S28**).

## 3. Models

### 3.1. TM5 Model

TM5 is a global chemistry transport model, which allows two-way nested zooming (Krol et al., 2005). In this study we apply the zooming with $1^o \times 1^o$ resolution over Europe, while the global domain is simulated at a horizontal resolution of $6^o$ (longitude) $\times 4^o$ (latitude). TM5 is an offline transport model, driven by meteorological fields from the European Centre for Medium-Range Weather Forecasts (ECMWF) Integrated Forecast System (IFS) ERA-Interim reanalysis (Dee et al., 2011). The spatial resolution of this data set is approximately 80 km (T255 spectral) on 60 vertical levels from the surface up to 0.1 hPa. We use 25 vertical layers (extending up to 0.2 hPa). The boundary layer, the free troposphere, and the stratosphere are represented by 5 (up to 1 km), 10, and 10 layers, respectively. The temporal resolution of the data is 3-hourly for near surface data (e.g., BLHs) and 6-hourly for 3D fields (temperature, wind and humidity).

Tracers in TM5 are transported by advection (in both horizontal and vertical directions), cumulus convection, and vertical diffusion. Tracer advection is based on the so-called "slopes scheme" which considers a tracer mass within a grid cell as a mean concentration and the spatial gradient of the concentration within the grid box (Russel and Lerner, 1981), which is caused by the motion of the tracer into and out of the grid box. Non-resolved transport by shallow cumulus and deep convection in TM5 is parameterized by a bulk mass flux approach originally described in Tiedtke (1989). Such convective clouds are described by single pairs of entraining/detraining plumes representing the updraft/downdraft motion. The parameterization of the vertical turbulent diffusion in the boundary layer is based on the scheme of Holtslag and Moeng (1991), while the formulation of Louis (1979) is considered in the free troposphere. The BLH is computed by using the expression of Vogelezang and Holtslag (1986), as described in Section 3.2. The



exchange coefficients from the vertical diffusion are combined with the vertical convective mass
fluxes to calculate the sub-grid scale vertical tracer transport. After redistributing the tracer mass
by convection and diffusion, the slopes are updated. Since in convective areas, transport in the
vertical can be more efficient than in the horizontal, van der Veen (2013) decreased the vertical
slopes (called "updated slopes treatment" in Section 4) through an adjustment scheme. The
author found an improvement of the inter-hemispheric mixing gradient in TM5, which was
initially underestimated as reported in e.g., Patra et al. (2011). This "updated slopes treatment"
has been used for the sensitivity tests described below. Furthermore, we performed sensitivity
tests using directly the convection fields from the ECMWF IFS model, instead of the default
convection scheme based on Tiedtke (1989). The ECMWF convection scheme includes several
improvements of the parameterizations of deep convection, radiation, clouds and orography,
introduced operationally since ECMWF ERA-15 analyses (e.g., Gregory et al., 2000; Jakob and
Klein, 2000; Morcrette et al., 2001).  Finally, we evaluate the combination of the "updated slopes
scheme" and the convection scheme based on ECMWF.

3.2. TM5 Boundary layer height scheme

Vertical mixing in the atmospheric boundary layer is mostly turbulent. The BLH is confined by a
thin vertical layer where steep vertical gradients of pollutants, trace gases, and aerosol occur.
Consequently, all the observational devices built for the retrieval of BLH are based on the search
of the height at which the strongest gradients occur. These gradients can be in either the
atmospheric potential temperature profile, the wind profile, or the aerosol backscatter profile. For
meteorological and atmospheric transport models, the bulk Richardson number, a dimensionless
parameter defined as the ratio between the buoyant consumption by thermal stability and the
mechanic generation by wind shear, has been widely used to determine BLHs (e.g., Vogelezang
and Holtslag, 1986; Seibert et al., 2000; Seidel et al., 2012). Thus, BLH is the vertical level at
which the bulk Richardson number ($R_{ib}$) computed from the ground reaches a critical value $R_{ic}$
characterizing the passage of turbulent fluid flow to laminar one. In the TM5 model, the
expression of Vogelezang and Holtslag (1986) is used to compute $R_{ib}$, as follows:

$$R_{ib} = \left(\frac{g}{\theta_{vS}}\right) \frac{(\theta_{vh} - \theta_{vS})(h - z_s)}{(u_h - u_S)^2 + (v_h - v_S)^2 + bu_*^2} \qquad (1)$$

where $g$ is the gravitational acceleration (9.81 m s$^{-2}$), $h$ the geopotential height of the model, $\theta_{vs}$
the virtual potential temperature at the surface and $\theta_{vh}$ the virtual potential temperature at the
model level $h$. $z_s$ corresponds to the surface geopotential height. $u_s$ denotes the zonal wind speed
at the surface and $u_h$ the zonal wind speed at the model level $h$. $v_s$ denotes the meridional wind
speed at the surface and $v_h$ the meridional wind speed at the model level $h$. $bu_*^2$ depicts the
turbulence production due to the surface friction, a term which also prevents an undetermined $R_{ib}$
in case of uniform high wind speeds relevant for neutral boundary layers. $b$ is a coefficient
determined to be 100 (Vogelezang and Holtslag, 1986) and $u_*$ is the surface friction velocity.
The geopotential heights h and z$_s$ are expressed in m. The potential temperature is in K and the
velocities are in m/s.

The vertical profile of $R_{ib}$ is linearly interpolated from the first layer of the model until $R_{ib}$
reaches its critical value $R_{ic}$. Commonly, a $R_{ic}$ value of 0.25 has been used (e.g., Vogelezang and



Holtslag, 1986; Seibert et al., 2000; Seidel et al., 2012) while in TM5 a $R_{ic}$ value of 0.3 has been
applied. Moreover, the minimum BLH in TM5 is set to 100 m.
3.3 InGOS $^{222}$Rn flux map
We use the new $^{222}$Rn flux map developed by Karstens et al. (2015) within the InGOS project
(called hereafter 'InGOS $^{222}$Rn flux map'). This map is based on a parameterization of $^{222}$Rn
production and transport in the soil, using a deterministic model based on the equations of
continuity and diffusion (Fick's 1$^{st}$ law) to compute the transport of the $^{222}$Rn flux from the soil
to the atmosphere. The modelled radon flux is dependent on soil porosity and moisture, with the
latter obtained from two different soil moisture data sets, i.e., from the Land Surface Model
Noah (driven by NCEP-GDAS meteorological reanalysis), and from the ERA-Interim/Land
reanalysis, respectively. In this study we apply the $^{222}$Rn flux map version based on the Noah soil
moisture data set. Furthermore, the $^{222}$Rn flux map considers the water table (from a hydrological
model simulation), the distribution of the $^{226}$Ra content in the soil, and the soil texture. For
comparison, we apply also the commonly used constant emission maps with uniform continental
$^{222}$Rn exhalation of 21.98 mBqm$^{-2}$s$^{-1}$ between 60°S and 60°N; uniform continental $^{222}$Rn
emissions of 11.48 mBqm$^{-2}$s$^{-1}$ between 60°N and 70°N (excluding Greenland); and zero flux
elsewhere (Jacob et al., 1997). The InGOS $^{222}$Rn flux map provides monthly $^{222}$Rn fluxes over
the 2006-2011 period, aggregated to 0.5°×0.5° grid for Europe and complemented by the
constant emissions for the regions outside Europe. **Figures 3a** and **3b** illustrate the spatial and
mean seasonal variations of the $^{222}$Rn fluxes from the InGOS $^{222}$Rn flux map over Europe. The
modelled $^{222}$Rn flux is found to be larger in the areas where the $^{226}$Ra activity concentration in
the upper soil is very high, such as the Iberian Peninsula, areas in Central Italy and the Massif
Central in Southern France (**Figure 3a**). The mean seasonal variations of the $^{222}$Rn fluxes are
mainly driven by the soil moisture. On average, the InGOS $^{222}$Rn emissions over Europe are
smaller than the constant emission (except July - September; **Figure 3b**).
**4. Simulation setup**
4.1. Model boundary layer heights
We extract the TM5 BLHs using either the TM5 default expression of $R_{ib}$ (Section 3.2),
representing the effective BLH in the TM5 simulations, or based on Seidel et al. (2012) used in
the InGOS model validation exercise (i.e., $R_{ic} = 0.25$ and both surface wind and friction velocity
are set to zero in Eq.1; see Section 3.2). Furthermore, because InGOS and IGRA sites are not co-
located, we extract the BLH in the model both at the location of the InGOS station and at the
location of the nearest IGRA station, resulting in a set of four different modeled BLHs labelled
by the following acronyms:
• 'TM5': TM5 default version (Eq.1 in Section 3.2 with $R_{ic} = 0.3$); extracted at InGOS
stations by using 2D interpolation
• 'TM5_IGRA': As 'TM5', but extracted at IGRA station, which is closest to the selected
InGOS station





- 'TM5_INGOS': BLHs computed in TM5 model adopting the InGOS definition of the BLH (i.e., $R_{ic} = 0.25$ and both surface wind and stress velocity are set to zero in Eq.1), extracted at InGOS station.
- 'TM5_INGOS_IGRA': As 'TM5_INGOS', but extracted at IGRA station, which is closest to the selected InGOS station

Furthermore, we evaluate the BLHs as provided by ECMWF analyses and interpolated to TM5 grids (labelled 'ECMWF'). The values of these BLHs are extracted only at the InGOS stations. The ECMWF BLH is determined using an entraining parcel method, selecting the top of stratocumulus, or cloud base in shallow convection situations (Dee et al., 2011).

4.2. Simulated $^{222}$Rn activity concentrations

We simulate $^{222}$Rn activity concentrations using either the InGOS $^{222}$Rn flux map based on Noah soil moisture data, or constant $^{222}$Rn fluxes (see Section 3.3). Furthermore, we apply four different convection schemes in the TM5 model (for the InGOS $^{222}$Rn flux map based simulations only). These different simulations are labelled by the following acronyms:
- FC_CT: constant $^{222}$Rn fluxes, and default convection scheme in TM5 based on Tiedtke (1989)
- FI_CT: InGOS $^{222}$Rn flux map, and default convection
- FI_CS: InGOS $^{222}$Rn flux map and updated treatment of slopes in the TM5 convection scheme (see Section 3.1)
- FI_CE: InGOS $^{222}$Rn flux map and the updated convection scheme based on ECMWF reanalyses (see Section 3.1).
- FI_CU: InGOS $^{222}$Rn flux map, updated treatment of slopes and updated convection scheme based on ECMWF

**5. Results**

5.1. Simulated boundary layer heights versus observations

We focus the analysis on the InGOS stations (measuring $CH_4$ and $N_2O$, and / or $^{222}$Rn activity concentrations; **Figure 1**) at low altitudes (i.e., excluding mountain stations) and compare the modelled BLHs with observations at the closest IGRA stations. **Figures 4** and **5** show the mean seasonal variation for the nocturnal (00 UTC) and daytime (12 UTC) BLH, respectively (2006-2010 average). The nocturnal BLHs show a clear seasonal cycle at most stations, with typically higher nocturnal BLHs during winter (but also larger range between 25% and 75% percentile) compared to summer. This seasonal pattern is very consistent between measurements and model simulations. However, at some continental stations (e.g. Heidelberg, Gif-sur-Yvette) the IGRA data show very low nocturnal BLHs (median value below 100m) during summer, which are not reproduced by the models (in particular not by the TM5 default BLH, which has an algorithmic-internal lower limit of 100m). In general, the Whisker plots (**Figure 4**) show a skewed (non-normal) distribution for most monthly data (observations and model simulations) with the median value being usually significantly lower than the mean. The daytime BLHs show a very pronounced seasonal cycle at most continental stations (opposite in phase with the seasonal cycle





of the nocturnal BLH), with typical values around 500m during winter, and ~1000-2000m during summer. The daytime BLH is in general relatively well simulated at most stations, as further illustrated by the ratios between modelled and observed BLHs, which are close to 1 (see **Figure S13** in the Supplement). An exception, however, are coastal sites (e.g., Angus, Mace Head), where apparently the model representation errors (e.g., transition between land and sea) are a limiting factor. In general, it should be expected that the model BLH extracted at the location of the IGRA station should agree better than that extracted at the InGOS station (See Section 4.1 for the definition of the model BLHs). However, e.g. at Egham the opposite is the case, since the IGRA station (Herstmonceaux) is closer to the coast, and the corresponding model BLH has more 'marine' character (and the transition zone between sea and land is not resolved by the model). For most 'non-coastal' sites, however, the difference between the BLH at the InGOS station and the IGRA station, as well as the difference between the TM5 default and 'TM5_INGOS' BLH is usually very small (**Figures 4 and 5 and Figures S12 and S13** in the Supplement). The ECMWF BLH is in some cases slightly different compared to the TM5 or 'TM5_INGOS' BLH, especially at coastal sites, probably partly also due to model-representation errors (different horizontal grids of the ECMWF IFS model and TM5 (see Section 3.1), and different methods of BLH computation (see Section 4.1)). Compared to the data for the nocturnal BLH, the daytime BLHs show much smaller difference between median and mean value, indicating a less skewed frequency distribution (**Figures S12 and S13** in the Supplement).
In the supplement (**Figures S2** to **S11**) we show the full time series for the 10 stations in 2009, illustrating that also the synoptic variability of the BLH is relatively well reproduced by the models (for both nocturnal and daytime BLH). Furthermore, we extend the analysis by using all IGRA stations over Europe (about 130 stations; see **Figure 1** and **Figures S14** and **S15** in the Supplement). This extended analysis confirms the major findings discussed above, especially (1) the relatively good agreement between simulated and observed BLH during daytime, (2) the tendency for the simulated nocturnal BLHs to be too high during summer, and (3) larger differences between TM5 and IGRA BLHs for stations located in costal zones.

In the following we include the ceilometer and lidar derived BLH at Cabauw and Trainou, respectively, in the analysis. As clearly visible from the correlation plot between ceilometer and IGRA data for Cabauw (**Figure 2**), the ceilometer BLHs during midday are usually lower than the IGRA data (especially for the period March to September), while modelled BLHs fall in between the two observational datasets (**Figure 6**). Part of this difference is likely due to the different methodologies. Hennemuth and Lammert (2006) pointed out that inconsistencies between the atmospheric thermal profile and the aerosol concentration profile can result in differences between radiosonde and lidar/ceilometer BLH retrievals. In addition, also the spatial separation between Cabauw and DeBilt (~23 km) combined with different surface characteristics (wetter soils in Cabauw and different large scale surface roughness) may play some role. While the correlation between IGRA BLHs and the ceilometer BLH retrievals at Cabauw is reasonable (r=0.63) during daytime (**Figure 2**), it is very poor during night (**Figure S1**), probably due to the issues of ceilometers to detect the shallow nocturnal BLH, as mentioned in Section 2.1.2. The lidar daytime data at Trainou for 2011 agree relatively well with the model BLHs (except May) (**Figure 7**). While no IGRA data are available for this period, the comparison between model simulations and IGRA for 2006-2010 at Trainou (**Figure 5**) shows similar (or slightly better) agreement as the comparison between lidar and model for 2011.



1  5.2. Simulated $^{222}$Rn activity concentrations versus observations

**Figures 8** and **9** show the mean seasonal variations of observed and simulated $^{222}$Rn activity
concentrations at each of the studied InGOS sites at 05 UTC (time around which typically the
daily maximum $^{222}$Rn activity concentration occurs) and at 14 UTC ($^{222}$Rn daily minimum),
respectively. For most stations, TM5 simulated $^{222}$Rn activity concentrations based on the InGOS
$^{222}$Rn flux map show significantly better agreement with observations than the simulations based
on the constant $^{222}$Rn flux, especially regarding the average seasonal variations. The
improvement is largest during winter months, when TM5 simulations based on the constant
$^{222}$Rn fluxes often overestimate observations, while simulated concentrations based on the
InGOS $^{222}$Rn flux map are significantly lower owing to the lower $^{222}$Rn fluxes (**Figure 3b**). This,
in turn, is driven mostly by the higher soil moisture and consequently lower permeability of the
soil in winter. Furthermore, large differences are visible at many North European sites close to
the coast (Angus, Lutjewad, Mace Head, Cabauw), where the water table can be very shallow,
significantly reducing the $^{222}$Rn fluxes (Karstens et al., 2015). Apparently, model simulations
based on the InGOS $^{222}$Rn flux map (which include modelled water table in the parameterization
of $^{222}$Rn fluxes) agree much better with observations than the control runs with constant $^{222}$Rn
fluxes. Despite the larger $^{222}$Rn fluxes during summer, daily minimum $^{222}$Rn concentrations in
model and observations are usually lower at continental stations (e.g. Heidelberg, Gif-sur-
Yvette) due to the much higher daytime boundary layer in summer compared to winter.
**Figures S18 to S28** in the supplement show the full time series of simulated and observed $^{222}$Rn
concentrations at the 10 studied InGOS stations (with $^{222}$Rn observations available) for 2009.

In the following, we analyze the relationship between $^{222}$Rn activity concentration and BLH in
more detail. **Figure 10** shows the mean seasonal diurnal cycle of observed and simulated $^{222}$Rn
activity concentration and BLH for the four seasons at different sites. The figure illustrates the
very strong anti-correlation between simulated BLH and $^{222}$Rn activity concentration: The
modelled BLHs increase sharply between 9:00 and 10:00 UTC (10:00/11:00 and 11:00/12:00
LT), resulting in an immediate decrease of modelled $^{222}$Rn concentrations. In contrast, the $^{222}$Rn
activity concentration measurements show a slower decrease over several hours. Apparently the
sharp changes in the 'model world' are due to the relatively coarse temporal resolution of
ECMWF meteorological data (3-hourly for surface data (e.g., BLHs) and 6-hourly for 3D fields
(temperature, wind and humidity); see Section 3.1). Because the ceilometer data at Cabauw
during night might be questionable, we included in **Figure 10** only the lidar measurements at
Trainou (TR4) that shows a much slower growth of the BLH, starting in the morning and
reaching its maximum in the late afternoon, as also illustrated in Pal et al. (2012, 2015). In spite
of the obvious issue of the temporal resolution of the model, **Figure 10** illustrates that the
mismatch between simulated and observed $^{222}$Rn activity concentrations cannot be explained by
the modeled BLH. Especially during daytime, the TM5 BLHs are close to the IGRA
measurements at most stations, while larger differences are observed between $^{222}$Rn activity
concentration simulations and measurements at several stations. This is further illustrated in
**Figure 11**, where we compare the ratio of simulated and observed BLH with the ratio of
simulated and observed $^{222}$Rn activity concentration during daytime, and in **Figure 12** where
these ratios are shown for the different seasons. This finding points to potential shortcomings of
TM5 to correctly simulate the vertical $^{222}$Rn activity concentration gradients within the boundary
layer (see below). Furthermore, it is important to consider the uncertainties of the $^{222}$Rn flux





map. Karstens et al. (2015) estimated that the most important uncertainty in the $^{222}$Rn flux is due to the uncertainties in the soil moisture data. Altogether, the uncertainty in modelled $^{222}$Rn fluxes for individual pixels ($0.083^o \times 0.083^o$) are estimated to about 50%. Karstens et al. (2015) pointed out that the uncertainty of the $^{222}$Rn fluxes averaged over the footprint of the measurements might be smaller. However, the uncertainties of neighboring pixels in the $^{222}$Rn flux map are likely strongly correlated, and therefore the reduction of the relative uncertainty (integrated over a typical footprint on the order of 50-200km) is probably relatively small. Assuming an overall uncertainty of ~50% of the regional $^{222}$Rn fluxes, the model simulations could be considered broadly consistent with observations at most sites.

The use of the new ECMWF based convection combined with updated treatment of slopes (i.e., FI_CU acronym in Section 4.2) results in a small decrease of simulated $^{222}$Rn concentrations at most stations, typically on the order of ~10-30% (see **Figures S31 to S41** in the Supplement). However, root mean square (RMS) and correlation coefficients are very similar at most sites for both convection parameterizations (**Figure 11**). Hence, no clear conclusions can be drawn, which parameterization is more realistic. At the same time, **Figure 11** demonstrates again the improvement using the InGOS $^{222}$Rn flux map, resulting in (1) ratios between simulated and observed $^{222}$Rn activity concentration closer to one, (2) lower RMS, and (3) higher correlation coefficients at several stations, compared to the model simulations using constant $^{222}$Rn fluxes. This highlights the challenge to validate model simulations. The difference of ~10-30% of $^{222}$Rn activity concentrations using a different convection parameterization is expected to result in a difference of similar order of magnitude for the GHG emissions derived in inverse modelling. First GHG inversions with the new ECMWF based convection confirm that derived emissions change significantly (not shown). **Figure 12** illustrates further that the ratio between observed and simulated daytime $^{222}$Rn activity concentration also depends on the exact hour, decreasing significantly between 12:00 and 15:00 UTC at several stations (very pronounced at Trainou and Ispra). This is clearly due to the shortcomings of TM5 to simulate the diurnal cycle in the BLH discussed above (owing to the coarse temporal resolution of the meteorological data). In the current TM5-4DVAR system the average (observed and simulated) concentrations between 12:00 and 15:00 LT are used to derive emissions (Bergamaschi et al., 2010; 2015). Given the too fast increase of the BLH and consequently too fast decrease of simulated mixing ratios in the morning transition period, the choice of the assimilation time window may introduce some systematic errors in the flux inversions.

In the analyses shown in **Figure 12**, the data include all meteorological conditions. In addition, we performed this analysis separately for unstable, neutral, and stable vertical mixing conditions, based on the bulk Richardson number near the surface in TM5 model. This extended analysis, however, showed relatively similar model performance for these different weather conditions (results not shown).

Finally, we explore the vertical gradients of TM5 simulated $^{222}$Rn activity concentrations at Cabauw where measurements are available at two vertical levels (20 m [CB1] and 200 m height [CB4]. The tower height of 20 m is within the first model layer, while 200 m is within layer 3. **Figure 13** shows the mean diurnal cycle of modeled and observed vertical gradients of $^{222}$Rn activity concentrations for each month for 2009. Although the InGOS $^{222}$Rn flux based model simulations agree better with observations (in terms of $^{222}$Rn activity concentrations; see **Figure**



**8 and 9**) compared to the model simulations based on constant fluxes, this is not the case for the $^{222}$Rn gradients for some months (between June and November the modelled gradients based on the constant fluxes agree better with observations). During large parts of the year, the InGOS $^{222}$Rn flux based model simulations underestimate the observed gradients. This is further illustrated in the scatter plots shown in **Figure 14** (separately for 00 and 12 UTC). For inverse modelling, especially the underestimated vertical gradient during daytime is critical and could lead to biases in the GHG inversions. Furthermore, **Figures 13** shows that during the transition phase in the morning the modelled $^{222}$Rn activity concentration vertical gradient decreases faster than the observed gradient, which is again probably largely due to the coarse time resolution of the meteorological data in TM5, but could point in addition also to too fast vertical mixing in the model.

## 6. Conclusions

In the first part of this study, we evaluated the boundary layer dynamics of the TM5 model by comparison with BLHs from the NOAA IGRA radiosonde data as well as with BLH retrievals from a ceilometer at Cabauw and lidar at Trainou.
TM5 reproduces reasonably well the IGRA BLHs during daytime within 10-20% (which is within the uncertainty of the IGRA data) for continental stations at low altitudes. During night, the model overestimates the shallow nocturnal BLHs, especially for very low BLHs (<100 m) observed during summer time. At coastal sites, the differences between simulated BLH and IGRA data (both day and nighttime) are usually larger due to model representation errors (since the transition zone between the marine boundary layer over sea and the continental boundary layer over land is not resolved by the model).
The BLH retrievals at Cabauw show a moderate correlation with IGRA data from De Bilt at 12 UTC, but are systematically lower. During night (00 UTC), however, the two data set show only a very poor correlation. Besides the fundamental differences in the BLH retrieval methods, however, also the spatial separation between Cabauw and DeBilt (~23 km) probably contributes to the differences in the derived BLH. For the lidar BLH data from Trainou, no direct comparison with the IGRA data is available (due to different time periods), but the comparison with the modelled BLH show similar agreement with the two different observational datasets [IGRA: for 2006-2010; lidar: 2011]. For the better exploitation of ceilometer / lidar data in the future, the further development of BLH retrievals is essential to ensure consistency between the different methods.

In the second part of this study, we compared TM5 simulations of $^{222}$Rn activity concentrations with quasi-continuous $^{222}$Rn measurements from 10 European monitoring stations.
The $^{222}$Rn activity concentration simulations based on the new $^{222}$Rn flux map show significant improvements compared to $^{222}$Rn simulations using constant $^{222}$Rn fluxes, especially regarding the average seasonal variability and generally lower simulated $^{222}$Rn activity concentrations at North European sites close to the coast. These improvements highlight the benefit of the process-based approach, including a parameterization of water table (Karstens et al., 2015). Nevertheless, the (relative) differences between simulated and observed daytime minimum $^{222}$Rn concentrations are larger for several stations (on the order of 50%) compared to the (relative) differences between simulated and observed BLH at noon. This is probably partly related to the





uncertainties in the $^{222}$Rn flux map (estimated to be on the order of 50%). In addition, however,
also potential shortcomings of TM5 to correctly simulate the vertical $^{222}$Rn activity concentration
gradients are likely to play a significant role, which may be caused by the vertical diffusion
coefficients and/or the limited vertical resolution in the model.
The comparison of simulated $^{222}$Rn activity concentrations with measurements at Cabauw (20 m
versus 200 m) shows that the model underestimates the measured vertical gradient (i.e.,
differences of concentrations between 20m and 200m levels) at this station. Furthermore, the
current coarse temporal resolution of the TM5 meteorological data (3-hourly for surface data and
6-hourly for 3D fields) limits the capability of simulating the diurnal cycle realistically. The
sharp increase of the modeled BLH in the morning transition period results in a rapid decrease of
the simulated $^{222}$Rn activity concentrations, while $^{222}$Rn measurements show a slower decrease at
many stations. This issue probably leads to systematic biases in inversions of GHG emissions.
An updated TM5-4DVAR system is currently under development with increased temporal
resolution of the meteorological data (3-hourly ECMWF data, interpolated to observational data
time).
Finally, we evaluated the updated slopes treatment and the new ECMWF based convection
scheme in the TM5 model. The results show a relatively small impact of the new slopes
treatment, but a significant impact of the new ECMWF convection scheme, leading to
significantly lower $^{222}$Rn activity concentrations (about 20%) during daytime, especially in
winter. While this is expected to have a significant impact on derived emissions in GHG
inversions, the comparison with the available European $^{222}$Rn activity concentration observations
showed very similar performance. Hence, no clear conclusion about which parameterization is
more realistic can be drawn from this study.

**Code availability**

28  Further information about the TM5 code can be found at http://tm5.sourceforge.net/. Readers
29  interested in the TM5 code can contact Maarten Krol (maarten.krol@wur.nl), Arjo Segers
30  (arjo.segers@tno.nl) or Peter Bergamaschi (peter.bergamaschi@jrc.ec.europa.eu)
31

**Acknowledgment**:

This work has been supported by the European Commission Seventh Framework Programme
(FP7/2007–2013) project InGOS under grant agreement 284274. We thank Juha Hatakka for
providing $^{222}$Rn data from Pallas. Furthermore, we are grateful to Clemens Schlosser from the
German Federal Office for Radiation Protection for the $^{222}$Rn data from Schauinsland, which
were used for additional analyses. ECMWF meteorological data has been preprocessed by
Philippe Le Sager into the TM5 input format. We are grateful to ECMWF for providing
computing resources under the special project 'Global and Regional Inverse Modeling of
Atmospheric $CH_4$ and $N_2O$ (2012-2014)' and 'Improve estimates of global and regional $CH_4$ and
$N_2O$ emissions based on inverse modelling using in-situ and satellite measurements (2015-
2017)'.



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





**Table 1:** Description of the different surface stations measuring $^{222}$Rn activity concentrations. See Figure 1 for the locations of the stations shown by their ID

| Station ID | Name | Country | Latitude (o) | Longitude (o) | Altitude/Height (m) | $^{222}$Rn instrument | Reference |
|---|---|---|---|---|---|---|---|
| PAL | Pallas | Finland | 67.97 | 24.12 | 572/7 | one-filter method | Hatakka et al. (2003) |
| TTA | Angus | United Kingdom | 56.55 | -2.98 | 363/50 | ANSTO | Smallman et al. (2014) |
| LUT | Lutjewad | Netherlands | 53.40 | 6.35 | 61/60 | ANSTO | van der Laan et al. (2010) |
| MHD | Mace Head | Ireland | 53.33 | -9.90 | 40/15 | one-filter method | Biraud et al. (2000) |
| CBW (CB1) | Cabauw | Netherlands | 51.97 | 4.93 | 19/20 | one-filter method | Vermeulen et al. (2011) |
| CBW (CB4) | Cabauw | Netherlands | 51.97 | 4.93 | 199/200 | ANSTO | Vermeulen et al. (2011) |
| EGH | Egham | United Kingdom | 51.43 | -0.56 | 70/10 | one filter method | Levin et al. (2002) |
| GIF | Gif-sur-Yvette | France | 48.71 | 2.15 | 167/7 | one-filter method | Lopez et al. (2012), Yver et al. (2009) |
| HEI | Heidelberg | Germany | 49.42 | 8.71 | 146/30 | one-filter method | Levin et al. (2002) |
| TRN (TR4) | Trainou | France | 47.95 | 2.11 | 311/180 | ANSTO | Schmidt et al. (2014) |
| IPR | Ispra | Italy | 45.80 | 8.63 | 223/3.5 (15)[1] | ANSTO | Scheeren and Bergamaschi (2012) |

[1]measurements at 3.5m 'normalized' to sampling height of 15m based on wind-speed dependent correction (see Section 2.2)





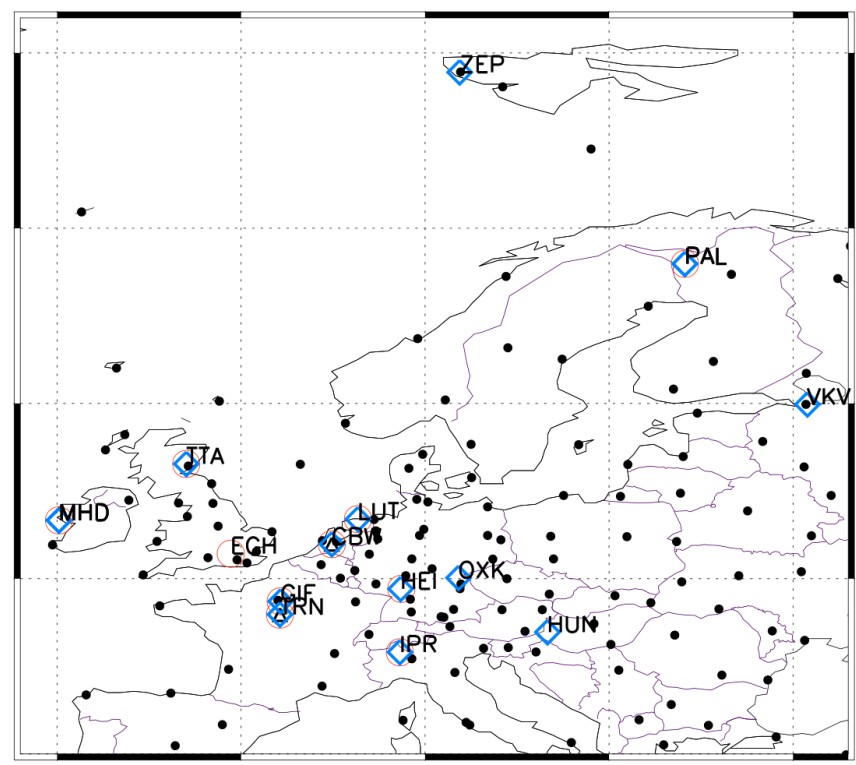

**Figure 1**: Observational network of InGOS greenhouse gas ($CH_4$, $N_2O$) and radon ($^{222}Rn$) concentration measurements and boundary layer height observations., Blue diamonds (◊):INGOS stations that measure $CH_4$ and/or $N_2O$ concentrations; red circles (○):InGOS stations that measure radon ($^{222}Rn$) activity concentrations; black dots (●): IGRA stations; triangles (Δ):ceilometer/lidar measurement sites (i.e., Cabauw/Trainou).



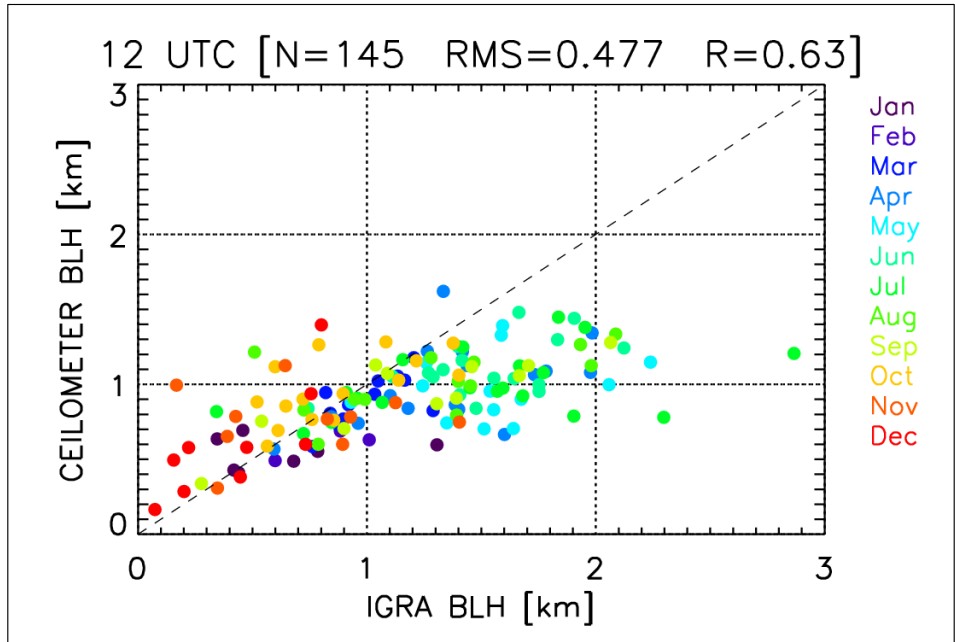

**Figure 2:** The Cabauw ceilometer boundary layer heights versus IGRA (De Bilt station) data for the year 2010 at 12 UTC are shown. The statistics (RMS in km and correlation coefficient R) are indicated as well as the number of pair of data (N) used to compute these metrics. The different colors indicate the months at which the data were obtained




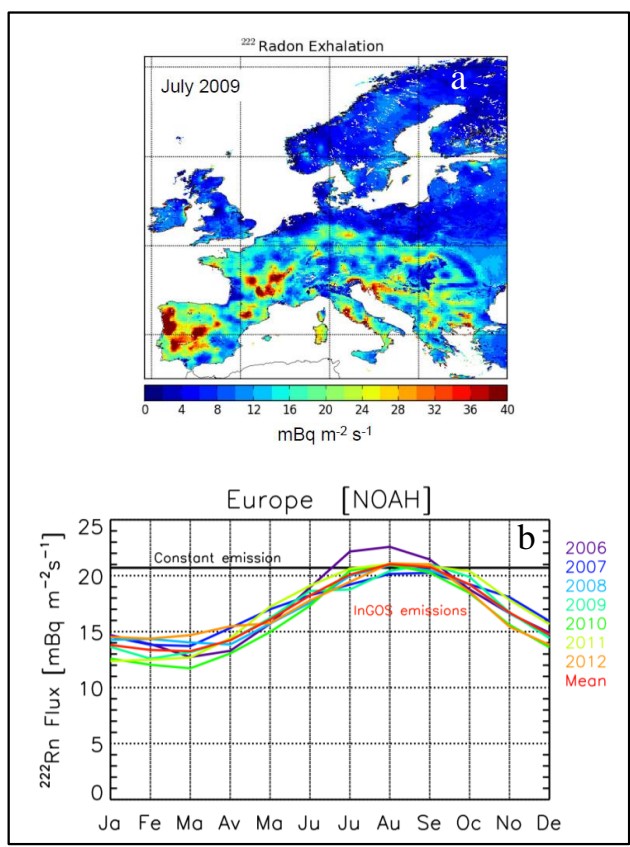

**Figure 3**: Radon ($^{222}$Rn) emissions used for the model simulations. (a) spatial distribution of InGOS emissions over Europe during July 2009. (b) seasonal and inter-annual variations of InGOS emissions (in different colors for different years; mean in red) and the commonly used constant emissions (black). The mean seasonal variations are averaged over the geographic domain between 10°W and 30°E longitude and between 35°N and 70°N latitude.



**Figure 4:** Observed (IGRA) and modelled (TM5, TM5_IGRA, TM5_INGOS, and TM5_INGOS_IGRA, ECMWF) boundary layer heights for InGOS stations at 00 UTC (2006-2010). The titles of each panel show the names and acronyms of the InGOS station, and the names of the nearest IGRA station used for comparison. The Whisker plots show the monthly minimum and maximum values (bars), and the 25% and 75% percentiles (boxes). The median values are given by the horizontal line and the mean values by the open circles in the boxes. The IGRA data are in blank and the various colors represent the various boundary layer heights from the TM5 and ECMWF models. The different acronyms of the model data are defined in Section 4 of the text

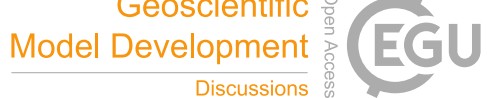

**Figure 4**: continued





**Figure 5**: As Figure 4, but at 12 UTC





**Figure 5**: continued





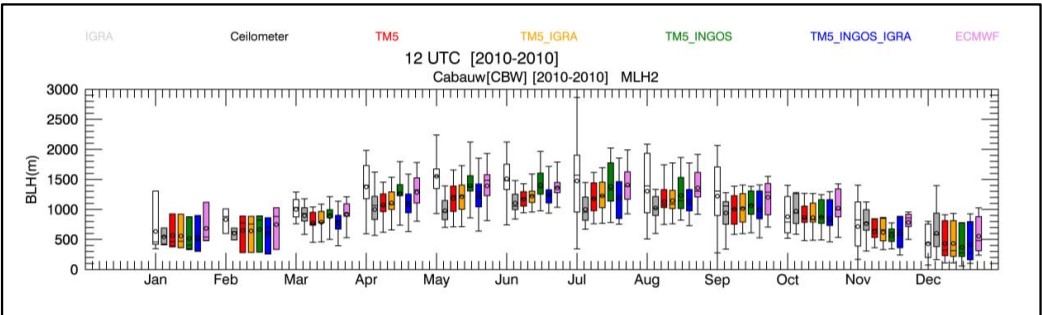

**Figure 6**: As Figure 4, but at Cabauw (CBW) where both ceilometer and nearby IGRA data (from de Bilt) are available. Observed and simulated boundary layer heights at 12 UTC and for 2010 are shown. IGRA data and ceilometer data are shown in blank and dark grey, respectively. The model data are represented by the colored boxes. The different acronyms are defined in Section 4 of the text



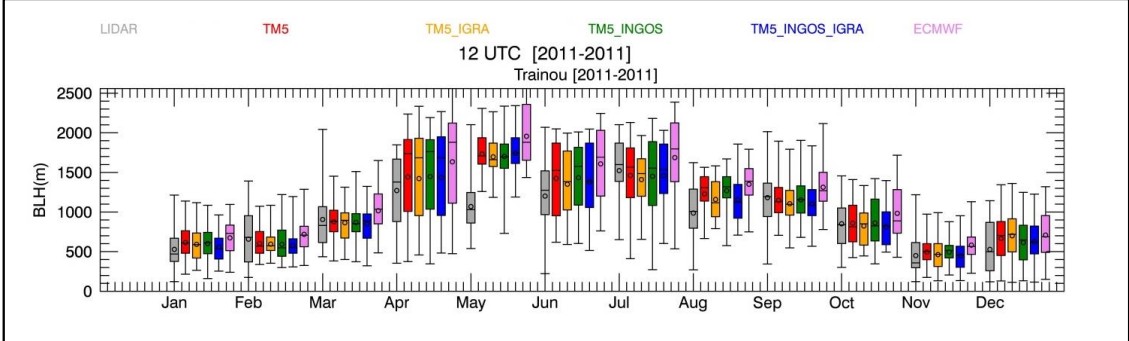

**Figure 7**: As Figure 4, but at Trainou (TRN) where lidar data are available during 2011. Lidar and model boundary layer heights at 12 UTC are shown. The lidar data are in dark grey. The model data are represented by the colored boxes. The different acronyms are defined in Section 4 of the text.







**Figure 8:** Seasonal variations of daily maximum of observed and simulated radon ($^{222}$Rn) activity concentrations at InGOS sites at 05 UTC (2006-2011). The Whisker plots show the monthly minimum and maximum values (bars), and the 25% and 75% percentiles (boxes). The median values are given by the horizontal line and the mean values by the open circles in the boxes. The observed radon activity concentrations are shown in black, and the model simulations are represented by the colored boxes (the different acronyms are defined in Section 4.2). FC uses constant $^{222}$Rn fluxes and FI the InGOS flux map.



**Figure 9**: As for Figure 8, but at 14 UTC illustrating the seasonal variations of daily minimum of radon ($^{222}$Rn) activity concentrations.




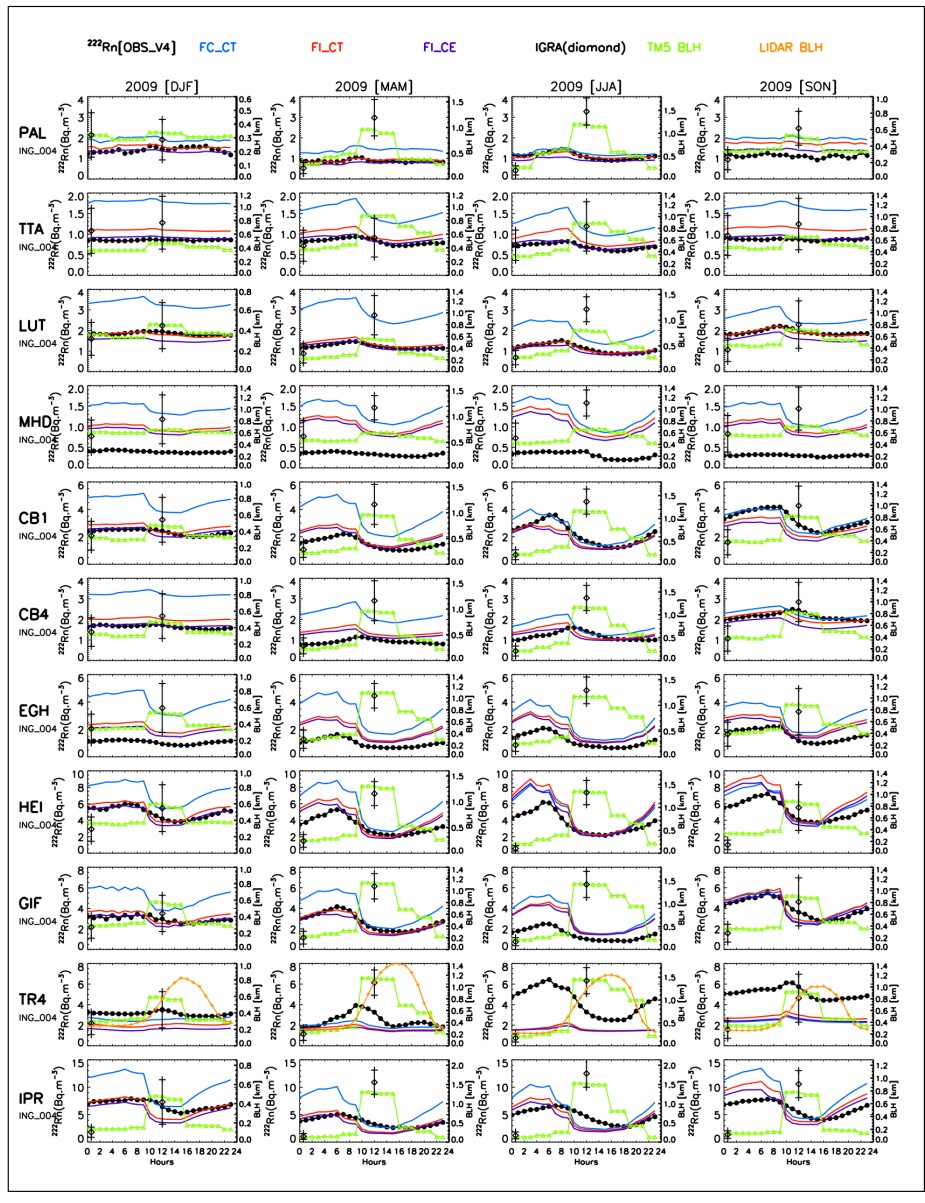

**Figure 10**: Seasonal variations of radon activity concentrations and boundary layer heights (BLH) at the InGOS stations that measure radon activity concentrations. The observed radon activity concentrations are represented by the black solid line with dots. Three model simulations are considered: FC_CT, the model simulations using constant emissions, FI_CT using the InGOS emissions and the default convection scheme of TM5; FI_CE using the the InGOS emissions and the new ECMWF convection scheme. The BLHs of TM5 are in green, while observed IGRA BLHs at 00 and 12 UTC are shown by the diamonds together with their uncertainties. The lidar BLHs at Trainou (for 2011) are shown by the solid orange line





**Figure 11**: Left: statistics of observed vs. simulated radon activity concentrations for the different stations (12 UTC). Right: statistics of observed (IGRA (●) and ceilometer (CEIL)/LIDAR (*)) vs. simulated boundary layer heights. (12UTC). The acronyms of the stations (x-axis) are given in Table 1. For the median and RMS values, the unit of the y-axis is given on the top of the relevant graphs. The different model settings are given on the top of the graphs. The number of pair of data for each station is larger than 500





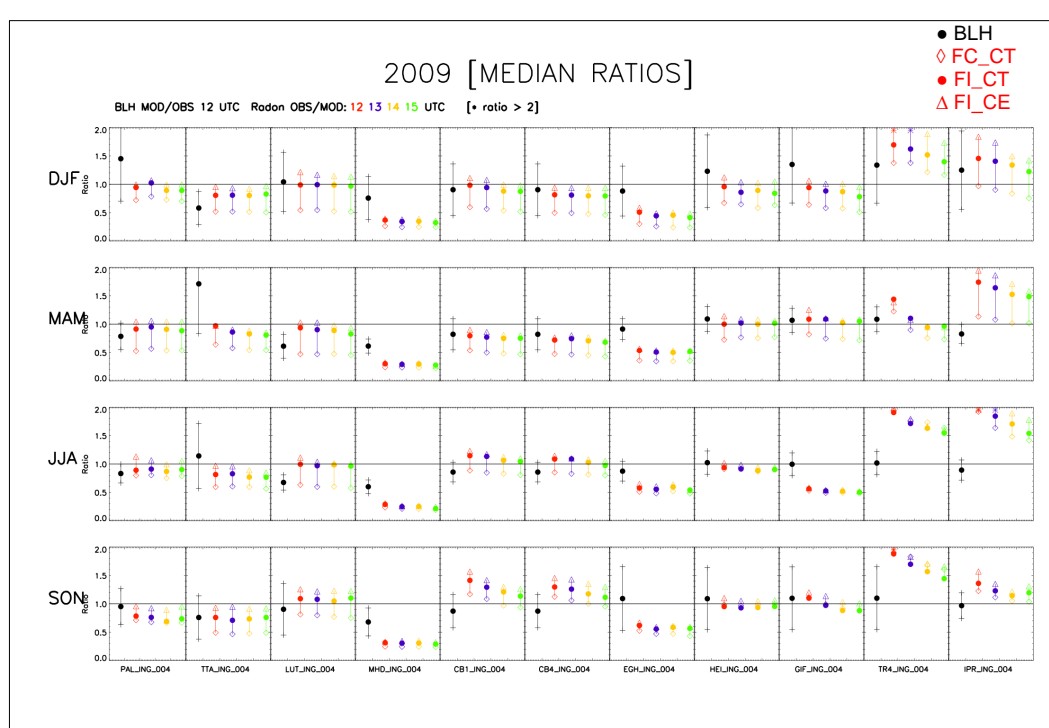

**Figure 12:** The seasonal variations of the ratios of BLHs (TM5/IGRA; black dot with solid line) at 12 UTC and the ratios of $^{222}$Rn activity concentrations (OBS/TM5) at 12, 13, 14, and 15 UTC for the 4 seasons [DJF; MAM; JJA; SON] of the year 2009 and for each InGOS radon measurement sites. The closest IGRA station to the radon measurement site is considered. Three TM5 simulations are shown here: The model simulations using the constant emissions [FC_CT; colored diamond], INGOS emissions and using the default convection scheme of TM5 (FI_CT; colored big dots) and using the new ECMWF convection scheme (FI_CE; colored triangles).

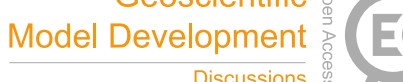



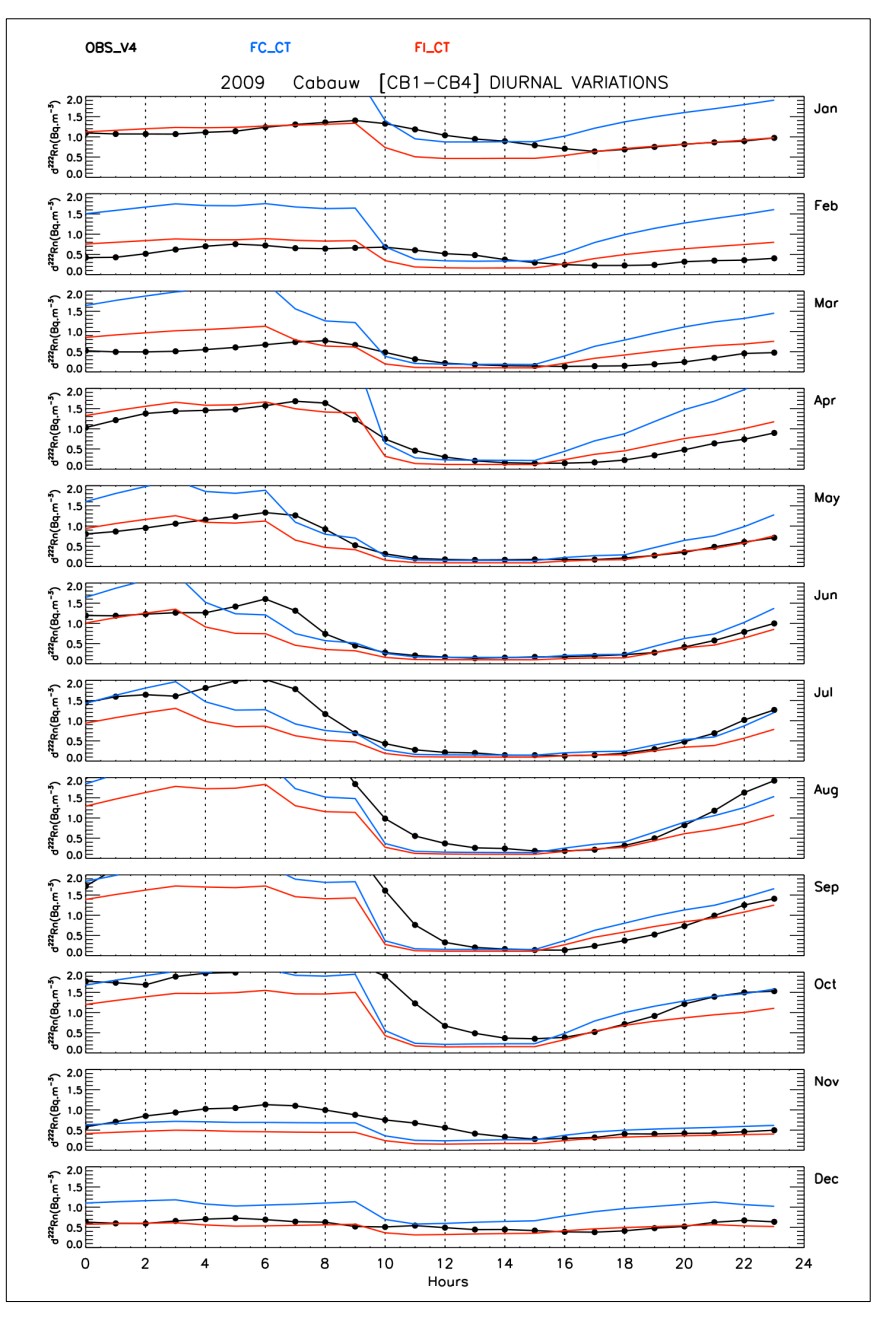

**Figure 13:** Mean diurnal variations of the radon activity concentration differences between the two measurement levels at Cabauw (20m, 200m). The observed gradient is shown by the black solid line with dots (for each month of the year 2009), and the modelled gradient by the solid blue line for the constant emissions (FC_CT) and by the solid red line for the INGOS emissions (FI_CT), respectively.





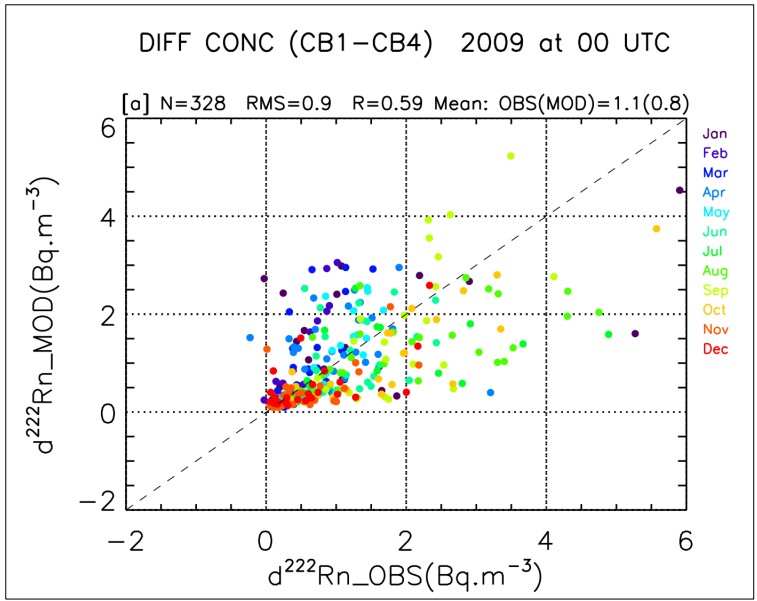

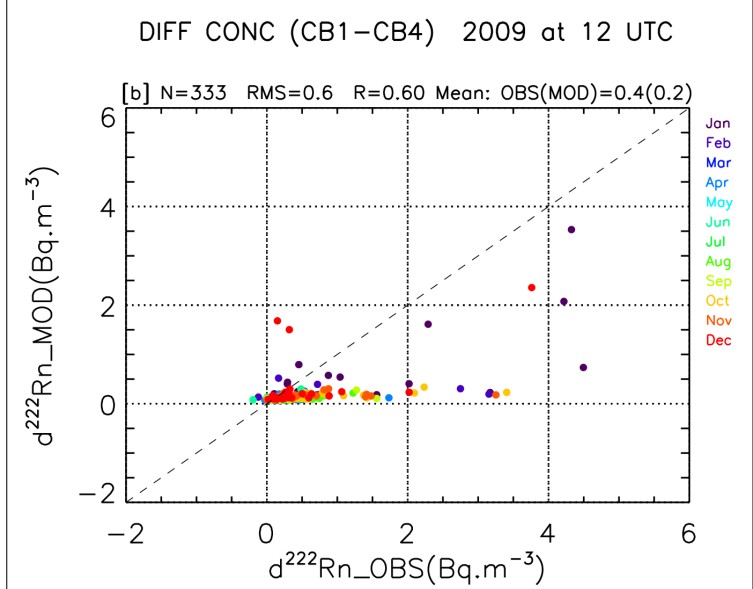

**Figure 14**: Correlation plots between the simulated and observed vertical [222]Rn gradients at Cabauw at 00 UTC (top) and 12 UTC (bottom). Model simulations using InGOS emissions (FI_CT termed as MOD) are shown. Each color indicates the month at which the data are obtained