# Peer review of "Evaluation of the boundary layer dynamics of the TM5 model over Europe"

_Geoscientific Model Development, 2016_

## Referee Comment (RC3)

The paper attempts to evaluate the performance of TM5 to simulate boundary layer heights and surface radon concentrations. Some biases are found that the authors link to some weaknesses in TM5. Overall, the paper is fairly well written but it is obvious that many people were involved in the analysis of data and model output which makes the paper appear 'fragmented' and, at times, unstructured and disorganized. Provided below are major and minor comments which also include some suggestions to improve the paper.

**Major comments**

1) Is Geosc. Model Dev. An appropriate journal for this type of paper? This paper addresses the evaluation of a model, not the development. A journal such as Atmospheric Chemistry and Physics or Boundary Layer Meteorology seems more appropriate to me.

2) The title is too broad and should be made more focused on those aspects that are actually studied in the paper, i.e. daytime and nocturnal boundary depths and 222Rn-concentration. Boundary layer dynamics include the study of thermodynamical and dynamical processes in the boundary layer including e.g. winds, stability, entrainment, etc. These processes are not studied in this paper and the title is therefore misleading. The title should reflect that the analysis is only made over Europe.

3) The difficulty of a coarse model to represent a coastal zone has not only to do with the coarseness of the model, but also the horizontal spatial variability. Also in high resolution models the largest spatial variability for fluxes can be found in these regions. For CO2 this has been addressed by Pillai et al. (2010).

4) There are some problems in the structure of the paper, and titles of sections are sometimes inappropriate/misleading. Also the introduction of some figures in the text is sometimes a bit strange. For example. Figure 2 is introduced very early, but is only discussed very late (much later than the discussion of Figs. 4 and 5). This must be resolved by either putting the discussion in the section where it is introduced, or before the model output is compared to observations. As for an example eof a misleading section titles, consider e.g. Section 4 which is entitled 'simulation setup'. Section 4.1 only addresses extraction of model output and no aspects of the simulation setup. These misleading/inappropriate titles should be corrected. It would be good to give subsection with appropriate titles in the Result section.

5) The ceilometer/lidar related part does not really fit in this paper. There are many issues with the comparability between radiosonde/lidar derived PBL heigths as discussed in many papers (and also obvious from Fig. 2) and you don't want to include these issues and uncertainties in this paper. In fact, including these data makes some conclusions in the paper rather weak. Figure 6 and 7 (and stars in Fig. 11) which include the ceilometer data do not add anything new and can easily be removed.

6) The authors mention coastal and non-coastal stations as well as mountainous stations (that they have removed from the evaluation). It would be nice to include the IGRA stations in table (not just Radon stations as is currently done) and indicate what stations are in coastal and mountainous regions. It also seems important that the authors explain how they define a coastal or mountainous station.

7) The reader is overwhelmed with data and figures (not to speak of the supplemental figures!). Reduce the number of figures and also the number of subfigures with certain figures. Some of this could be addressed by removing lidar/ceilometer related data as indicated in major comment 5. In Figs. 4 and 5, not all stations need to be shown. Just pick a few that clearly show some points you are making in the paper. It would also be nice to see in the figures which stations are in coastal/non-coastal terrain, as this seems important in the analysis (see previous comment on coastal and non-coastal stations).

**Minor comments**

- 1. P2, general: The abstract is very long (almost longer than the introduction) and reads like a summary.
- 2. P2, line 4: "dynamics" should be "height"
- 3. P3, line 15: define BLH properly, is it above the surface (depth) or above sea level (height).
- 4. P4, line 11: Section title could also be depth, depending on definition
- 5. P4, line 19: The equation of bulk Richardson number should be introduced here and not on page 7.
- 6. P4, lines 19-22: There should be some more explanation on choices made and how to use the bulk Richardson number. For example, how is theta\_v calculated from IGRA-soundings? The neglection of u\* is hardly explained, but this is stressed in the Seidel 2012-paper, a citation here would help.
- 7. P5, line 13: The introduction of this figure is very strange, as it is not discussed here.
- 8. Figure 2: Including the ceilometer data is not recommended as mentioned in the major comments. We see here clearly one of the issues in that ceilometer is underestimating blh from IGRA. A complicated issue that is not suitable for the current paper.
- 9. P6, line 5: unclear: +/- 10 to +/- 15% ? or does +/- means approximately?
- 10. P6, line 9: 15m inlet should with a space. The paper has many of these types of typos. Please check.
- 11. P6, section 3.1: the addition of a figure where vertical resolution of TM5 model and radiosonde are compared would be helpful. This would also make clear at what exact depths the TM5 model gives output. Then, as an example one could examine a typical boundary layer depth in this figure. Keep in mind that many readers of Geos. Mod. Dev. are probably not familiar with a concept like boundary layer height. See also major comment on appropriateness of journal.
- 12. P6, line 30: there are 60 vertical levels below 0.1 hPa and 25 layers below 0.2 hPa. How dense is the layering between 0.1 and 0.2 hPa? Or is it ECMWF and TM5 layering?
- 13. P7, line 5 . The idea of an "updated slopes scheme (treatment?)" is very unclear and should be clarified.
- 14. P7, line 19: Delete "vertical". "aerosol" should be plural.
- 15. P7, line 20: All the observational devices...... are based on the search..." Not an accurate statement. For example, sometimes strongest gradients occur right at the surface.
- 16. P7, line 21: "can be either" should be "can be based either on".
- 17. P7, line 42: m/s is m s-1.
- 18. P7, line 44: Unclear/ambiguous sentence.
- 19. P8, line 1: Why is a value of Ri\_c of 0.3 used in TM5 and not the more common value of 0.25? Should be an easy fix for the model developers.
- 20. P8, line 8 and 14: What is the difference between '222Rn flux map' and the 'InGOS 222Rn flux map' one? Be sure that the 'abbrevations' are used properly throughout the text.
- 21. P8, line 18: mBqm-2s-1. Some spaces are lacking in the unit.
- 22. P8, line 30-32: How can the extraction (or calculation) of variables (model boundary layer heights) be a simulation set-up. See also one of the major comments.

- 23. P8, line 42: What does 2D interpolation exactly mean? Various 2D approaches exist. Be specific and more accurate here.
- 24. P8, section 4.1: Is it really valuable to have so many different definitions? Besides, in this section, I would expect some discussion about the representation of the grid points chosen with respect to reality of the stations as this seems important for your discussion later on (coastal and non-coastal).
- 25. P9, line 7: ECMWF can be added as a bullet point.
- 26. P9, section 4.2: it is very unclear what type of simulations have been done. Consider a table.
- 27. P9, line 31: for clarity, at least one bl-profile with the different calculations of bl-height could be shown. Here, also vertical resolution of both IGRA and models can be shown. Besides, you can point out the differences generally found for a nocturnal and daytime (a 00 and 12 UTC) bl- figure, for example.
- 28. P9, line 34: Which mountain stations, and how did you define a mountain station? You could add labels in Table 1. The same holds for coastal and non-coastal stations, it is not defined what they are, this could be labeled in Table 1 as well.
- 29. P10, line 4: "coastal sites". Why don't you show a map with the representation of these two stations in the several data points extraction?
- 30. P10, line 11: How are non-coastal sites defined?
- 31. P10, line 15: "probably". What makes you think probably and not certainly?
- 32. P10, line 25: "relatively" compared to what? And are you surprised by these results? It is well known that Sbls are very shallow, and often these are missed by the model anyway.
- 33. P10, line 27: costal should be coastal.
- 34. P10, line 31: As mentioned in previous comments, figure 2, and, in general, ceilometer related data, should be removed in this paper (the correlation is poor and subject to many discussions that are not appropriate to discuss in this type of paper). Furthermore, this figure that shows observations vs. observations does not fit in a section which is called simulations vs. observations?
- 35. P10, line 37: DeBilt should be De Bilt.
- 36. P10, Figures 6 and 7 are redundant, see one of the major comments.
- 37. p10, lines 29 to 45 should be removed. See previous comments on ceilometer data.
- 38. P11, line 4-5: About the timing of Rn-concentrations (05 and 14 UTC). Does this hold for both summer and winter?
- 39. P11, line 14: The list of coastal stations gets longer throughout the paper. Add it as labels in the paper. Is Cabauw really coastal station? I don't think locals would agree.
- 40. P11, line 24: This could be the start of an additional section.
- 41. P11, line 30: What do you mean by "Apparently"?
- 42. P11, line 31: What's a "model world"?
- 43. P11, line 44: "This finding..." What finding exactly?
- 44. P9-13, Section 5. what about a station selection for the figures? You seem to overwhelm the reader with graphs and bars, whereas only few things are to be highlighted. For example, the coastal and non-coastal zones are interesting, some are necessary to show due to later analysis with the Rn- and BLH combination. But certainly not all the stations are necessary. Then space would be saved, figures could be enlarged, these would be better readable and the article in general would be better appreciated. All the other redundant stations can then be stored in the supplemental material (which is very large as well).
- 45. P10-11, Section 5: Can be better divided in more sections.
- 46. P12, line 11: Add section 5.4 for this paragraph.
- 47. P12, line 38: "weather conditions", I suppose you mean "stability regimes"?

- 48. P12, line 42-43: About CB1 and CB4, see remarks about table.
- 49. P13, line 14 (section 6): this section feels more like a summary than a conclusion.
- 50. P13, line 16: "dynamics" should rather be "height". The dynamics are not evaluated.
- 51. P13, line 19: 10-20%, this is the first time I see a statistical value between TM5 and observations. That's very late.
- 52. P13, line 23: IGRA observations (not "data").
- 53. P13, line 26: moderate correlation or reasonable? In any case, it is not good and opens up the floor for many discussions that are not relevant to the topic of the paper. See comments before about removing ceilometer related analysis.
- 54. P13, lines 26-35: remove ceilometer related analysis/results.
- 55. P14, line 23-24: It is indeed difficult to draw conclusions. Try to be more quantitative, perhaps add more statistics, and this would make it easier to draw conclusions.
- 56. Table 1. Extend this table with labels as coastal and non-coastal stations. what is ANSTO? What is CB1? CB4? Probably different levels at the tower? The average readers do not have previous knowledge of the dataset and an attempt needs to be made to make it more readable for them. Maybe you want to change CB1 and CB4 to level 1 and level 2. Are the 'o' for latitude and longitude degree (°) signs? What does Altitude/Height exactly mean? Do you mean Terrain elevation (Mean Sea Level) and Height (Above Ground Level), respectively?
- 57. Figure 1. What are the abbrevations? The authors should refer to Table 1. Some letters are very hard to read, consider another color for either the names, or for the black dots. The black triangles and orange circles are barely visible. Where are the coastal stations exactly? What are the vertical and horizontal lines? Longitude and latitude? It should be indicated on the axis.
- 58. Figure 2. remove. See major/minor comments
- 59. Figure 3. Vertical and horizontal axis in the upper diagram?
- 60. Figure 4. These figures are very small. Can the whisker plots be centralized around the months to which they are concerning to? And maybe the spacing then between the whisker plots could be enlarged. The scaling on y-axis is not the same. It doesn't have to, but it should be stated in the caption. Although the scales are not very far apart, so probably same axis length would work. This is true for almost all figures.
- 61. In the text it is referred to coastal and non-coastal stations. It would be helpful to highlight that in the figure. Is it really necessary to show all stations? Maybe you could make coastal stations fig4a and continental stations in fig. 4b?
- 62. Figure 6/7. Redundant/remove
- 63. Figure 11: remove CEIL/LIDAR data from figure.
- 64. Figure 12. What is ratio? TM5 divided by IGRA?
- 65. Figure 13. Abbreviations are not explained well in caption (e.g what is CB1 and CB4?). "Mean diurnal variations..."
- 66. Figure 13: Data points are outside the y-axis range. This should be corrected.
- 67. Figure 14. How can the R of the lower figure be almost the same as the R for the upper figure?

**References:**

Pillai, D., Gerbig, C., Marshall, J., Ahmadov, R., Kretschmer, R., Koch, T., & Karstens, U. (2010). High resolution modeling of CO 2 over Europe: implications for representation errors of satellite retrievals. Atmospheric Chemistry and Physics, 10(1), 83-94.

---

## Referee Comment (RC1) · Anonymous Referee #1 · 29 Apr 2016

General comments

I think this paper could be significantly enhanced by including some further discussion or even recommendations on estimating model transport errors based on the model-observation comparisons of 222Rn and BLHs. As the authors already point out, transport errors are a substantial source of uncertainty in the fluxes estimated in atmospheric inversions. There are already a number of groups using TM5 in atmospheric inversions, but such recommendations need not be limited only to TM5 but in general the use of the new 222Rn emission map and the IGRA BLH dataset for assessing model transport errors.

The paper includes many detailed figures of the comparison of BLHs and 222Rn but I think a couple of figures that summarize (i.e. give a more immediate indication of)

the comparison between the model and observations and of the seasonal and diurnal cycles could be very helpful. Then some of the detailed figures could be moved into the supplement.

Specific comments

P3, L21: Here the authors mention only surface monitoring stations in regional inversions but not aircraft data, which are often used (e.g. the Kort et al. 2008 study cited here). Model representation of aircraft observations will be also affected by errors in BLH and simulations of boundary layer dynamics. Perhaps this should be mentioned.

P10, L26-27: It is interesting that the modelled nocturnal BLHs tend to be higher than observed in summer but that this is not the case in winter? Can the authors comment on this?

P10, L41: The authors do not discuss comparison of the modelled and observed (at IGRA sites) nighttime BLHs for Cabauw or Trainou.

P11, L7: Please give a quantitative estimate of "better agreement" either stating the improvement in the RMSE or correlation.

P11, L15: Please delete "apparently" – either the InGOS $222Rn$ flux maps give better agreement or they don't, so "apparently" is not appropriate here.

P11, L38-39: The authors state that the mismatch between the observed and modelled $222Rn$ activity concentrations cannot be due the modelled BLH because this matches the observed BLH well. However, I understand that the modelled BLH is determined by vertical interpolation, therefore, I wonder if the vertical resolution in TM5 may be a possible reason for the mismatch?

P13, L1-11: I think this section should be expanded to discuss the influence of compensating errors in the $222Rn$ fluxes (in the constant versus InGOS flux maps) and in the BLHs and how this might explain the fact that the simulations with the constant fluxes lead to a better comparison with the observations.

[Figure]

Technical comments

P4, L46: "as" should be replaced by "compared to"

P10, L36: delete "also" after "In addition".

---

## Referee Comment (RC2) · Anonymous Referee #2 · 2 May 2016

**General comments**

This study reports on a thorough evaluation of TM5 to describe the boundary layer dynamics, comparing various parameterization settings of the BL and extraction methods height to radiosonde, lidar and ceilometer observations. Furthermore simulations of 222Rn using two different emissions and various settings for advection and convection in TM5 are compared. The study draws potentially important conclusions regarding uncertainties due to convection parameterization in TM5, relevant for GHG emission studies, and is therefore well suited for publication in GMD.

While this study is certainly thorough, in its current shape the manuscript is merely a report on the numerous sensitivity runs that have been executed. A more rigorous selection of sensitivity experiments to be presented, along with a more selective presentation of observational data could largely improve the readability of the manuscript. Also the abstract is currently too elongated.

For instance, both presenting 'TM5' and 'TM5-IGRA' and likewise 'TM5-INGOS' and 'TM5-INGOS-INGRA' in figures 4-7 seems not necessary, as 'differences are usually very small' (p.10, l.13)' , and furthermore such differences cannot be explained in terms of sensitivity of the parameterization, but rather reflect a representation error. Therefore I believe these simulation results are even a bit confusing and should be removed from the figures.

In Figures 8 and 9 a clear improvement with the revised $222Rn$ emission map is visible, but differences between various convection/advection parameterizations is less obvious, which makes me wonder if presentation of all these results could not be more condensed, or moved to the supplementary material.

I believe the figures 4-9 benefit from presenting only seasonal mean statistics, rather than monthly means: The same messages can be conveyed with much condensed use of figures.

Also the authors put large emphasis on the improvement in the comparison to $222Rn$ observations when using the new flux map. However, the purpose of this paper is rather the evaluation of the boundary layer dynamics in TM5, by performing sensitivity runs. While many figures are presented, in the end it remains unclear to me how the parameterizations quantitatively compare, presented preferably in a Table. E.g. the statistics of the analyses given in Figures 11 and 12 could be averaged over the different stations, while excluding coastal stations hampered by representation errors and excluding the results obtained with the simplified flux map.

On the abstract, I believe the authors should condense this strongly, by reporting only the key findings of this study, which I believe are the performance of TM5 to represent BLH (l.14-l.17), and the achievements and limitations of the comparison against the new $222Rn$ flux map.

**Detailed comments**

**Abstract**

Please consider to condense especially lines 3-12. Also I suggest to remove the conclusion regarding the improvement with the new Karstens et al. emissions from the abstract because, even though interesting, it is not essential to the subject of this manuscript. Consider re-formulation of sentence on l. 21-24, which is difficult to grasp. Also lines 37-42 read a bit confusing: while ECMWF convection results in much lower 222Rn activity than TM5 the authors cannot conclude if this is an improvement or not, which in its current formulation, does not appear a useful finding.

**Introduction**

I expect a few more references to studies to previous work that have considered the relevance of boundary layer dynamics for trace gas distributions, (and inversions), e.g. Locatelli et al., GMD 2015. How does this new work relate to that study?

**Section 2**

Page 5, l. 14: You introduce a figure where you compare the Cabauw ceilometer BL with IGRA data. I expect some discussion and interpretation of this result at this point.

**Section 3**

Here, and at several points throughout the manuscript, you mention the issues associated to the resolution of TM5 (1x1 horizontally over Europe, 25 vertical layers, 3-hourly surface meteo data, 6-hourly 3D fields). Considering it's apparent relevance it would have been interesting to see a sensitivity study at different model resolutions. Could you specify the temporal resolution of the ECMWF convection fields in your sensitivity study? Is this 6 hour?

P 8, l 14: 'Noah soil moisture data' : Do you have a reference here? Considering it's apparent sensitivity to soil moisture, why didn't you consider use of the ERA-Interim

reanalysis? This would be more consistent with the 222Rn atmospheric model simulations, or?

**Section 4**

"We extract...": Are TM5 simulations of 222Rn collocated in time and space with respect to the observations? Please be more specific here on horizontal, vertical an time interpolation.

**Section 5.**

Pp 10, l 14 – l19: So do you have any indication that ECMWF treatment of the BLH is better than the one currently used in TM5, based on this? Please provide more quantitative conclusions.

Pp 11, l 38: "The mismatch (...) cannot be explained by the modelled BLH". This statement seems inconsistent with Figure 12, where 'potential shortcomings of TM5 to correctly simulate the vertical 222Rn activity concentration gradients' are illustrated. Please explain this apparent inconsistency.

P12, l3-l7: Karstens et al. pointed out that the uncertainty averaged over the footprint might be smaller than 50

P12, l22: the authors suggest that the GHG-emissions derived in inverse modeling change by the same order of magnitude as 222Rn, i.e. 10-30

P12, l35-39: Please provide a short interpretation of this sensitivity analysis.

P12, l43: "tower height of 20m is within the first model layer 200m is within layer 3.": Considering it's sensitivity, how did you treat the model sampling? Did you apply vertical interpolation? Do you expect any sensitivity to vertical model resolution?

P13, l11: In this section, and in Figure 13, I miss results from the FI-CE run using the ECMWF meteo. Or are differences marginal? Please comment.

[Figure]

Figure 11, right panels and Figure 12: Which parameterizations are used for the computation of the TM5 boundary layer height? Standard TM5 or ECMWF convection? Or is the difference in BLH for the two parameterizations marginal?

**Conclusions**

P 14, l17 "The updated slopes treatment": This is jargon. Please reformulate to something more generic, e.g. "the revised advection parameterization". Could you indicate the importance of this study for GHG inversions based on TM5? Is this study a ground for replacing the convection treatment in TM5, or is it merely useful in providing a constraint on the uncertainty estimate of the GHG emission inversions?
* * *

---

## Author Comment (AC1) · 27 Jul 2016

Please find 3 pdf files (put in a single zip file: see supplement file below) relative to our reply to the comments on the paper:

1. Our responses to the comments of the referee #1

3. The revised text and figures with tracked changes compared to the submitted version

4. The revised supplement

Please also note the supplement to this comment:
http://www.geosci-model-dev-discuss.net/gmd-2016-48/gmd-2016-48-AC1-supplement.zip

---

## Author Response (AR1)

We thank the reviewer for his/her constructive review. In what follows, the comments of the reviewer are in italic and our reply in normal face.

**General comments**

I think this paper could be significantly enhanced by including some further discussion or even recommendations on estimating model transport errors based on the model-observation comparisons of 222Rn and BLHs. As the authors already point out, transport errors are a substantial source of uncertainty in the fluxes estimated in atmospheric inversions. There are already a number of groups using TM5 in atmospheric inversions, but such recommendations need not be limited only to TM5 but in general the use of the new 222Rn emission map and the IGRA BLH dataset for assessing model transport errors.

We have added several recommendations at the end of the conclusions.

The paper includes many detailed figures of the comparison of BLHs and 222Rn but I think a couple of figures that summarize (i.e. give a more immediate indication of) the comparison between the model and observations and of the seasonal and diurnal cycles could be very helpful. Then some of the detailed figures could be moved into the supplement.

We have reduced the number of figures, and reduced the number of scenarios shown both for the comparison of the BLH and  $^{222}$ Rn activity concentrations (as described in more detail in reply to reviewers #2 and #3).

**Specific comments**

P3, L21: Here the authors mention only surface monitoring stations in regional inversions but not aircraft data, which are often used (e.g. the Kort et al. 2008 study cited here). Model representation of aircraft observations will be also affected by errors in BLH and simulations of boundary layer dynamics. Perhaps this should be mentioned.

We have modified the text, mentioning explicitly the use of aircraft data in the study of Kort et al. (2008). Furthermore, we added a reference (Miller et al., 2013), which also use aircraft data for their flux inversion.

P10, L26-27: It is interesting that the modelled nocturnal BLHs tend to be higher than observed in summer but that this is not the case in winter? Can the authors comment on this?

Obviously, the model has in particular difficulties to simulate the very shallow nocturnal BLH, which is often observed at continental stations in summer. This is partly due to the fixed lower limit of 100m for the BLH in the model (see Figures S12 and S13 in the revised version of the Supplement).

P10, L41: The authors do not discuss comparison of the modelled and observed (at IGRA sites) nighttime BLHs for Cabauw or Trainou.

We had not discussed in detail the comparison of modelled and observed nighttime BLHs at Cabauw and Trainou due to the limitations of the ceilometer / LIDAR measurements during night (see section 2.1.2).

*P11, L7: Please give a quantitative estimate of "better agreement" either stating the improvement in the RMSE or correlation.*

The statement refers primarily to Figures 8 and 9 (Figures 6 and 7 the revised version), which shows the seasonal variation of observed and simulated 222Rn activity concentration. In additional, the improvement is also clearly visible in the overall statistics, shown in Figure 11 (Figure 8 in the revised version), which shows the improvement both in the RMSE values and correlation coefficients. This is briefly discussed later (in Section 4.2 of the revised manuscript).

*P11, L15: Please delete "apparently" – either the InGOS 222Rn flux maps give better agreement or they don't, so "apparently" is not appropriate here.*

**Deleted as suggested**

P11, L38-39: The authors state that the mismatch between the observed and modelled 222Rn activity concentrations cannot be due the modelled BLH because this matches the observed BLH well. However, I understand that the modelled BLH is determined by vertical interpolation, therefore, I wonder if the vertical resolution in TM5 may be a possible reason for the mismatch?

The dependence of the TM5 BLH on the vertical resolution has not been investigated. However, we note that the TM5 BLH (evaluated in the model version with 25 vertical layers) is in general very close to the ECMWF ERA-Interim BLH (60 vertical layers).

P13, L1-11: I think this section should be expanded to discuss the influence of compensating errors in the 222Rn fluxes (in the constant versus InGOS flux maps) and in the BLHs and how this might explain the fact that the simulations with the constant fluxes lead to a better comparison with the observations.

This is a good point. We have added a short statement that this could point to partially compensating systematic errors (See Section 4.2 in the revised version).

Technical comments

**P4, L46: "as" should be replaced by "compared to"**

We have slightly modified the sentence to: "attribute the height of the residual layer of aerosol ... as height of the real mixed layer". The suggested "compared to" would change the content of the sentence

P10, L36: delete "also" after "In addition".

Deleted as suggested

We thank the reviewer for his/her constructive review. In what follows, the comments of the reviewer are in italic and our reply in normal face.

**General comments**

This study reports on a thorough evaluation of TM5 to describe the boundary layer dynamics, comparing various parameterization settings of the BL and extraction methods height to radiosonde, lidar and ceilometer observations. Furthermore simulations of 222Rn using two different emissions and various settings for advection and convection in TM5 are compared. The study draws potentially important conclusions regarding uncertainties due to convection parameterization in TM5, relevant for GHG emission studies, and is therefore well suited for publication in GMD.

While this study is certainly thorough, in its current shape the manuscript is merely a report on the numerous sensitivity runs that have been executed. A more rigorous selection of sensitivity experiments to be presented, along with a more selective presentation of observational data could largely improve the readability of the manuscript. Also the abstract is currently too elongated.

For instance, both presenting 'TM5' and 'TM5-IGRA' and likewise 'TM5-INGOS' and 'TM5-INGOS-INGRA' in figures 4-7 seems not necessary, as 'differences are usually very small' (p.10, l.13)', and furthermore such differences cannot be explained in terms of sensitivity of the parameterization, but rather reflect a representation error. Therefore I believe these simulation results are even a bit confusing and should be removed from the figures.

We have significantly reduced the number of sensitivity experiments shown in the main paper: For the TM5 boundary layer heights we show now in the revised version only the boundary layers heights evaluated with the InGOS definition (consistent with the definition used for the IGRA radiosondes), evaluated both at the InGOS stations and the adjacent IGRA stations (see Section 3.2 in the revised version). The additional evaluations of the BLH are now shown only in the supplementary material. For 222Rn activity concentrations, we show now only 3 cases (FC\_CT, FI\_CT, FI\_CU; see Section 3.4 in the revised version) in the main Figures. Also the abstract has been significantly shortened.

In Figures 8 and 9 a clear improvement with the revised 222Rn emission map is visible, but differences between various convection/advection parameterizations is less obvious, which

makes me wonder if presentation of all these results could not be more condensed, or moved to the supplementary material.

We have condensed the presentation of the various convection/advection parameterizations, and show now in the revised version only the simulations with the combined 'revised slopes scheme' and ECMWF ERA-Interim convection. Furthermore, we removed the paragraph on this issue from the abstract

*I believe the figures 4-9 benefit from presenting only seasonal mean statistics, rather than monthly means: The same messages can be conveyed with much condensed use of figures.*

We had deliberately chosen to show the monthly means and would like to keep this presentation, since it gives more detailed information (more precise representation of the seasonal evolution) than the seasonal means. As already explained above, to render more readable the different graphs, we show now in the revised version only the more relevant model experimental settings in these Figures.

Also the authors put large emphasis on the improvement in the comparison to 222Rn observations when using the new flux map. However, the purpose of this paper is rather the evaluation of the boundary layer dynamics in TM5, by performing sensitivity runs. While many figures are presented, in the end it remains unclear to me how the parameterizations quantitatively compare, presented preferably in a Table. E.g. the statistics of the analyses given in Figures 11 and 12 could be averaged over the different stations, while excluding coastal stations hampered by representation errors and excluding the results obtained with the simplified flux map.

Although the evaluation of the new 222Rn flux map is not the primary objective of this paper, realistic 222Rn emissions are an essential prerequisite for the model validation.

We prefer to keep the presentation of the statistics per station (Figure 11, now Figure 8 in the revised version), because of (1) considerable differences also among the non-coastal sites, and (2) the limited number of stations.

On the abstract, I believe the authors should condense this strongly, by reporting only the key findings of this study, which I believe are the performance of TM5 to represent BLH (l.14-l.17), and the achievements and limitations of the comparison against the new 222Rn flux map.

We have condensed the abstract significantly and deleted the paragraph on the different convection/advection parameterizations.

**Detailed** comments**

Abstract

See our reply above

Please consider to condense especially lines 3-12.

We shortened this part of the abstract.

Also I suggest to remove the conclusion regarding the improvement with the new Karstens et al. emissions from the abstract because, even though interesting, it is not essential to the subject of this manuscript.

We think that this conclusion is important because realistic 222Rn emissions are an essential prerequisite for the model validation. Therefore, we would like to keep this conclusion in the abstract.

Consider re-formulation of sentence on l. 21-24, which is difficult to grasp.

The sentence has been slightly rephrased

Also lines 37-42 read a bit confusing: while ECMWF convection results in much lower 222Rn activity than TM5 the authors cannot conclude if this is an improvement or not, which in its current formulation, does not appear a useful finding.

We have deleted this paragraph from the abstract.

**Introduction**

I expect a few more references to studies to previous work that have considered the relevance of boundary layer dynamics for trace gas distributions, (and inversions), e.g. Locatelli et al., GMD 2015. How does this new work relate to that study?

We have included the suggested reference Locatelli et al. (2015).

Section 2

Page 5, l. 14: You introduce a figure where you compare the Cabauw ceilometer BL with IGRA data. I expect some discussion and interpretation of this result at this point.

We have moved Figure 2 (submitted version) to the Supplement. The scatter plots of Cabauw ceilometer BLHs at 00 and 12 UTC are now shown in a single Figure S1 in the Supplement (and the figure caption updated accordingly).

Section 3

Here, and at several points throughout the manuscript, you mention the issues associated to the resolution of TM5 (1x1 horizontally over Europe, 25 vertical layers, 3-hourly surface meteo data, 6-hourly 3D fields). Considering it's apparent relevance it would have been interesting to see a sensitivity study at different model resolutions. Could you specify the temporal resolution of the ECMWF convection fields in your sensitivity study? Is this 6 hour?

The temporal resolution of the ECMWF ERA-Interim convective fields is 3 hours. However, in the TM5 version used in this study, 6 hourly 3D meteo fields were applied (See Section 3.1).

*P* 8, *l* 14: 'Noah soil moisture data' : Do you have a reference here?

The following reference of Rodell et al. (2004) has been added:

Rodell, M., P. R. Houser, U. Jambor, J. Gottschalck, K. Mitchell, C.-J. Meng, K. Arsenault, B. Cosgrove, J. Radakovich, M. Bosilovich, J. K. Entin, J. P. Walker, D. Lohmann, and D. Toll, 2004. The Global Land Data Assimilation System, Bulletin of the American Meteorological Society, 85(3): 381-394

Considering it's apparent sensitivity to soil moisture, why didn't you consider use of the ERA-Interim reanalysis? This would be more consistent with the 222Rn atmospheric model simulations, or?

Karstens et al. (2015) recommended the use of the new emission maps derived from the Noah reanalysis. The authors found that "comparison with observations suggests that the flux estimates based on the GLDAS Noah soil moisture model on average better represent observed fluxes". We included the conclusion from Karstens et al. (2015) in the text to explain our choice of the Noah data based 222Rn flux map.

Furthermore, we simulated the 222Rn activity concentrations also using the ERA-Interim based 222Rn flux map (not shown). These additional sensitivity runs showed overall poorer agreement with 222Rn observations that the Noah data based 222Rn simulations, confirming the conclusion of Karstens et al. (2015).

Section 4

"We extract. . . ": Are TM5 simulations of 222Rn collocated in time and space with respect to the observations? Please be more specific here on horizontal, vertical an time interpolation.

We apply 3 dimensional interpolation (i.e., horizontal and vertical interpolation) using the 222Rn activity concentrations of the neighboring grid cells. The model output are hourly averaged concentrations (which are directly compared to the hourly averaged observations)

Section 5.

Pp 10, l 14 – l19: So do you have any indication that ECMWF treatment of the BLH is better than the one currently used in TM5, based on this? Please provide more quantitative conclusions.

No. We only stated that the differences we sometimes observe between TM5 BLHs and ECMWF BLHs at some coastal sites may be attributed to i) the relatively finer spatial resolution of ECMWF (~80 km in horizontal on 60 levels) and ii) to the different treatment of BLH in the

two models. The ECMWF BLHs are not discussed anymore in the text, hence this sentence has been deleted

*Pp 11, 1 38: "The mismatch (. . .) cannot be explained by the modelled BLH". This statement seems inconsistent with Figure 12, where 'potential shortcomings of TM5 to correctly simulate the vertical 222Rn activity concentration gradients' are illustrated. Please explain this apparent inconsistency.*

This is maybe confusing. No, Figure 12 (submitted version; Figure 10 in the revised version) shows the ratios of boundary layer heights (modelled BLH versus observed BLH) at noon along with the ratios of 222Rn activity concentrations (observed versus simulated) at 12, 13, 14, 15 LT for different seasons. We found that at most of the studied stations, the modelled BLHs compare well with observed BLHs, while the differences between the simulated 222Rn activity concentrations and observed ones can be larger. This result points to potential shortcomings of TM5 to correctly simulate the vertical mixing of 222Rn activity concentrations within the boundary layer. The text has been updated.

*P12, 13-17: Karstens et al. pointed out that the uncertainty averaged over the footprint might be smaller than 50*

The uncertainty averaged over the footprint could be smaller. However, as discussed in the paper, the uncertainties of neighboring pixels in the 222Rn flux map are likely strongly correlated, and therefore the reduction of the relative uncertainty (integrated over a typical footprint on the order of 50-200km) is probably relatively small.

P12, l22: the authors suggest that the GHG-emissions derived in inverse modeling change by the same order of magnitude as 222Rn, i.e. 10-30

Yes, this is correct. As mentioned in the paper, this has also been confirmed by first GHG inversions with the new ECMWF based convection (not shown).

**P12, 135-39: Please provide a short interpretation of this sensitivity analysis.**

We analyzed the ratios of both boundary layer heights and 222Rn activity concentrations as shown in Figure 12 (submitted version; Figure 10 in the revised version) for the 3 main stability regimes (stable, neutral, unstable or fair). We used the modelled Richardson number obtained at the first level of the model to discriminate between the 3 stability regimes. Results for the three stability regimes are similar and similar to those obtained when considering sample covering all the stability regimes shown in Figure 10 (revised version). A limitation of this exercise was that for both stable and neutral stability regimes, we had at most stations, only few cases by seasons. The text has been revised

P12, 143: "tower height of 20m is within the first model layer 200m is within layer 3.": Considering it's sensitivity, how did you treat the model sampling? Did you apply vertical interpolation? Do you expect any sensitivity to vertical model resolution?

As mentioned above, 222Rn activity concentrations are 3-D interpolated, i.e. including vertical interpolation. Yes, we expect some dependence on the vertical resolution of the model. However, this has not yet been analyzed in detail. The 3-D interpolation is now stated (see Section 3.2)

P13, 111: In this section, and in Figure 13, I miss results from the FI-CE run using the ECMWF meteo. Or are differences marginal? Please comment.

We now present the simulations of 222Rn activity concentrations by using convection scheme based on ECMWF reanalysis (FI-CE) combined with the "revised slopes scheme (FI\_CS). The differences between FI-CT and FI-CS are marginal (Figures S14-S24 in the revised version of the Supplement), hence the differences between FI\_CT and FI\_CU are dominated by FI-CE

Figure 11, right panels and Figure 12: Which parameterizations are used for the computation of the TM5 boundary layer height? Standard TM5 or ECMWF convection? Or is the difference in BLH for the two parameterizations marginal?

In Figure 11 (submitted version), the TM5 default boundary layer was shown in the submitted version. We now show the boundary layer height extract at the closest IGRA station associated to the InGOS measurement sites (acronym TM5 INGOS IGRA; Figure 8 in the revised version)

**Conclusions**

*P* 14, 117 "The updated slopes treatment": This is jargon. Please reformulate to something more generic, e.g. "the revised advection parameterization".

We use now the term 'revised slopes scheme' throughout the paper.

Could you indicate the importance of this study for GHG inversions based on TM5? Is this study a ground for replacing the convection treatment in TM5, or is it merely useful in providing a constraint on the uncertainty estimate of the GHG emission inversions?

Since we did not find a significant difference / improvement of the 222Rn simulations with the new ECMWF convection, this study does not provide enough evidence, which would justify the replacement of the convection scheme. Further studies are currently performed within the TM5 modelling community (including the use of further tracers), however at this stage no clear conclusion can be drawn.

We thank the reviewer for his/her constructive review. In what follows, the comments of the reviewer are in italic and our reply in normal face

**Received and published: 3 June 2016**

The paper attempts to evaluate the performance of TM5 to simulate boundary layer heights and surface radon concentrations. Some biases are found that the authors link to some weaknesses in TM5. Overall, the paper is fairly well written but it is obvious that many people were involved in the analysis of data and model output which makes the paper appear 'fragmented' and, at times, unstructured and disorganized. Provided below are major and minor comments which also include some suggestions to improve the paper.

**Major comments**

1) Is Geosc. Model Dev. An appropriate journal for this type of paper? This paper addresses the evaluation of a model, not the development. A journal such as Atmospheric Chemistry and Physics or Boundary Layer Meteorology seems more appropriate to me.

Yes, we believe that GMD is appropriate:

(1) GMD lists under 'Aims and scope' explicitly 'full evaluations of previously published models', see:

http://www.geoscientific-model-development.net/about/aims\_and\_scope.html

(2) Moreover, the referee #2 explicitly stated that the paper is "well suited for publication in GMD".

2) The title is too broad and should be made more focused on those aspects that are actually studied in the paper, i.e. daytime and nocturnal boundary depths and 222Rn-concentration. Boundary layer dynamics include the study of thermodynamical and dynamical processes in the boundary layer including e.g. winds, stability, entrainment, etc. These processes are not studied in this paper and the title is therefore misleading. The title should reflect that the analysis is only made over Europe.

The paper investigates the boundary layer heights as well as the processes in the boundary layer (including dynamic and thermodynamic, etc...). When simulating the 222Rn activity concentration in TM5, all the thermodynamic and dynamic processes in the model are relevant. Thus, the evaluation of the model simulations of 222Rn activity concentrations implicitly includes the evaluation of the whole boundary layer dynamics.

We agree with the second statement regarding the focus of the paper on Europe and have updated the title accordingly: "Evaluation of the boundary layer dynamics of the TM5 model over Europe"

3) The difficulty of a coarse model to represent a coastal zone has not only to do with the coarseness of the model, but also the horizontal spatial variability. Also in high resolution models the largest spatial variability for fluxes can be found in these regions. For CO2 this has been addressed by Pillai et al.(2010).

We agree that also spatial variability of 222Rn fluxes close to the coast may also play a role for the simulation of 222Rn activity concentrations at stations close to coast. For the 222Rn fluxes, the largest effect should be related to the variability / gradient of the water table close to the coast. In principle, this should be covered by the 222Rn flux map (within the horizontal resolution of the input data sets)

4) There are some problems in the structure of the paper, and titles of sections are sometimes inappropriate/misleading. Also the introduction of some figures in the text is sometimes a bit strange. For example. Figure 2 is introduced very early, but is only discussed very late (much later than the discussion of Figs. 4 and 5). This must be resolved by either putting the discussion in the section where it is introduced, or before the model output is compared to observations. As for an example eof a misleading section titles, consider e.g. Section 4 which is entitled 'simulation setup'. Section 4.1 only addresses extraction of model output and no aspects of the simulation setup. These misleading/inappropriate titles should be corrected. It would be good to give subsection with appropriate titles in the Result section.

We have revised the structure of the paper. Figure 2 has been moved to the Supplement. Several titles have been updated and subtitles have been added.

5) The ceilometer/lidar related part does not really fit in this paper. There are many issues with the comparability between radiosonde/lidar derived PBL heigths as discussed in many papers (and also obvious from Fig. 2) and you don't want to include these issues and uncertainties in this paper. In fact, including these data makes some conclusions in the paper rather weak. Figure 6 and 7 (and stars in Fig. 11) which include the ceilometer data do not add anything new and can easily be removed.

Despite the discussed limitations (especially for the ceilometer at Cabauw during night), we consider the ceilometer/lidar data useful as they provide information about the dynamic evolution, which is not well resolved by the IGRA data with only 2 measurements per day.

6) The authors mention coastal and non-coastal stations as well as mountainous stations (that they have removed from the evaluation). It would be nice to include the IGRA stations in table (not just Radon stations as is currently done) and indicate what stations are in coastal and

mountainous regions. It also seems important that the authors explain how they define a coastal or mountainous station.

We indicate now the chosen IGRA stations (which are closest to the InGOS stations) in the updated Figure 1.

We consider stations as 'coastal' (in a strict sense) if they are located at the coast (as e,g, Mace Head). However, when comparing with model simulations, also the horizontal model resolution has to be taken into account. Thus, model representation errors (related to the land / sea gradients) arise, if the model grid cell, in which the station is located, covers also a significant sea fraction. For stations, for which this is relevant (but which are not coastal stations in a strict sense) we choose now the term 'close to the coast'

7) The reader is overwhelmed with data and figures (not to speak of the supplemental figures!). Reduce the number of figures and also the number of subfigures with certain figures. Some of this could be addressed by removing lidar/ceilometer related data as indicated in major comment 5. In Figs. 4 and 5, not all stations need to be shown. Just pick a few that clearly show some points you are making in the paper. It would also be nice to see in the figures which stations are in coastal/non-coastal terrain, as this seems important in the analysis (see previous comment on coastal and non-coastal stations).

We have reduced the number of figures both in the main paper and in the Supplement. Furthermore, we have reduced significantly the number of scenarios. However, we would like to show the full set of InGOS stations, since this paper also aims to support the further analysis of the GHG flux inversions (CH4, N2O) performed within the InGOS project (manuscripts in preparation). We consider the link between these studies very important, since potential systematic errors in the simulation of the BLH dynamics (discussed in the present paper) could directly translate into systematic errors in the derived fluxes.

**Minor comments**

1. P2, general: The abstract is very long (almost longer than the introduction) and reads like a summary.

The abstract has been revised (and significantly shortened)

2. P2, line 4: "dynamics" should be "height"

We would prefer to keep the term "dynamics" (see also our reply to reviewer's major comment (2)

3. P3, line 15: define BLH properly, is it above the surface (depth) or above sea level (height).

The BLH is defined with reference to surface elevation, and not to sea level (Seidel et al., 2012). This has now been added in section 2.1.

4. P4, line 11: Section title could also be depth, depending on definition

We prefer to keep the title "Boundary layer height"

5. P4, line 19: The equation of bulk Richardson number should be introduced here and not on page 7.

Yes. The paragraph has been moved to Section 2.1 and updated

6. P4, lines 19-22: There should be some more explanation on choices made and how to use the bulk Richardson number. For example, how is theta\_v calculated from IGRA-soundings? The neglection of u\* is hardly explained, but this is stressed in the Seidel 2012-paper, a citation here would help.

This part of the text has been revised and the paper of Seidel et al. (2012) quoted again there

7. P5, line 13: The introduction of this figure is very strange, as it is not discussed here.

This Figure has been moved to the Supplement as Figure S1

8. Figure 2: Including the ceilometer data is not recommended as mentioned in the major comments. We see here clearly one of the issues in that ceilometer is underestimating blh from IGRA. A complicated issue that is not suitable for the current paper.

See our reply to the reviewer's major comments (5)

9. P6, line 5: unclear: +/- 10 to +/- 15%? or does +/- means approximately?

We have corrected this to '10-15%'

10. P6, line 9: 15m inlet should with a space. The paper has many of these types of typos. Please check.

The GMD convention seems to be not to use a space before 'm' (meter)

http://www.geoscientific-model-development.net/for\_authors/manuscript\_preparation.html

11. P6, section 3.1: the addition of a figure where vertical resolution of TM5 model and radiosonde are compared would be helpful. This would also make clear at what exact depths the TM5 model gives output. Then, as an example one could examine a typical boundary layer depth in this figure. Keep in mind that many readers of Geos. Mod. Dev. are probably not familiar with a concept like boundary layer height. See also major comment on appropriateness of journal.

We agree that such a figure would be useful. However, it would increase the number of figures that the reviewer asked to decrease. We refer the reader to the paper of Seidel et al. (2012), where the method is nicely illustrated in their Figure 1.

12. P6, line 30: there are 60 vertical levels below 0.1 hPa and 25 layers below 0.2 hPa. How dense is the layering between 0.1 and 0.2 hPa? Or is it ECMWF and TM5 layering?

The 25 vertical layers of the TM5 model version used in this study are defined as a subset of the 60 vertical layers of the ECMWF ERA-Interim reanalysis. The text has been updated.

13. P7, line 5. The idea of an "updated slopes scheme (treatment?)" is very unclear and should be clarified.

We have updated the short description of the "revised slopes scheme" (and use this term now throughout the text). For further details the reader is referred to van der Veen (2013).

14. P7, line 19: Delete "vertical". "aerosol" should be plural.

Has been corrected as suggested (moved to section 2.1)

15. P7, line 20: All the observational devices...... are based on the search..." Not an accurate statement. For example, sometimes strongest gradients occur right at the surface.

This should exclude indeed the gradients right at the surface.

16. P7, line 21: "can be either" should be "can be based either on".

Corrected as suggested

17. P7, line 42: m/s is m s-1.

Corrected as suggested

18. P7, line 44: Unclear/ambiguous sentence.

First, we computed the Richardson number Rib at each of the model levels by using the equation (1). To determine the boundary layer height, the vertical profile of Rib is interpolated linearly between consecutive levels. The BLH is defined as the height, where  $R_{ib}$  reaches the critical value Ric. The text has been updated.

19. P8, line 1: Why is a value of Ri\_c of 0.3 used in TM5 and not the more common value of 0.25? Should be an easy fix for the model developers.

Ric of 0.3 is the default value for the BLH determination in TM5, but there is no publication about this specific aspect. However, as discussed in Seidel et al. (2012), the choice of Ric close to 0.25 does not introduce large uncertainty. Moreover, the differences between BLHs determined by using Ric of 0.25 and Ric of 0.3 are very small (see e.g., Figures S2-S11 in the Supplement; acronyms TM5 and TM5\_InGOS).

20. P8, line 8 and 14: What is the difference between '222Rn flux map' and the 'InGOS 222Rn flux map' one? Be sure that the 'abbrevations' are used properly throughout the text.

It is the same flux map. This has been clarified throughout the text by using "InGOS 222Rn flux map"

21. P8, line 18: mBqm-2s-1. Some spaces are lacking in the unit.

In accordance with our response above, we keep it as it

22. P8, line 30-32: How can the extraction (or calculation) of variables (model boundary layer heights) be a simulation set-up. See also one of the major comments.

The extraction of the BLH according to the INGOS definition required some specific modification of the TM5 source code.

The text has been updated and "Simulation set up" has been deleted

23. P8, line 42: What does 2D interpolation exactly mean? Various 2D approaches exist. Be specific and more accurate here.

It is linearly interpolated between the grid cell and its closest neighbor (both along longitude and latitude). The text has been updated

24. P8, section 4.1: Is it really valuable to have so many different definitions? Besides, in this section, I would expect some discussion about the representation of the grid points chosen with respect to reality of the stations as this seems important for your discussion later on (coastal and non-coastal).

As already mentioned in our reply to the reviewer's major comments (item 7), we now consider only two definitions: "TM5\_INGOS" and TM5\_INGOS\_IGRA" that use the same expression of Bulk Richardson number, as performed for the IGRA data. TM5\_INGOS stands for the boundary layer heights (BLH) extracted at InGOS stations, while TM5\_INGOS\_IGRA is the BLH of the closest IGRA stations. The other model experimental settings are now defined in the Supplement and the relevant results are also shown in the Supplement

25. P9, line 7: ECMWF can be added as a bullet point.

This has been deleted in the main paper and put in the Supplement.

26. P9, section 4.2: it is very unclear what type of simulations have been done. Consider a table.

As already described above, now we use only two definitions of boundary layer height in the paper. The figures have been revised accordingly and are more readable

27. P9, line 31: for clarity, at least one bl-profile with the different calculations of bl-height could be shown. Here, also vertical resolution of both IGRA and models can be shown. Besides, you can point out the differences generally found for a nocturnal and daytime (a 00 and 12 UTC) bl-figure, for example.

As mentioned above, we refer the reader here to Seidel et al. (2012) (and their Figure 1, which clearly illustrates the method). Finally, we do not think that illustrating the vertical bl-profiles for both nocturnal and daytime will help in our discussions on IGRA BLHs and modelled BLHs

28. P9, line 34: Which mountain stations, and how did you define a mountain station? You could add labels in Table 1. The same holds for coastal and non-coastal stations, it is not defined what they are, this could be labeled in Table 1 as well.

For mountain stations, we excluded InGOS measurement sites such as e.g., Jungfraujoch and Schauinsland in the analysis. About the coastal sites, see our reply to the reviewer's major comments (item 6)

29. P10, line 4: "coastal sites". Why don't you show a map with the representation of these two stations in the several data points extraction?

We now show the locations of the IGRA stations associated to InGOS in Figure 1.

Regarding "coastal sites" see our reply to the reviewer's major comments (item 6)

*30. P10, line 11: How are non-coastal sites defined?*

We have not defined objectively "coastal or non-coastal stations" as explained above or in our responses to the reviewer's major comments in item 6

31. P10, line 15: "probably". What makes you think probably and not certainly?

This sentence has been deleted when revised the text

32. P10, line 25: "relatively" compared to what? And are you surprised by these results? It is well known that Sbls are very shallow, and and often these are missed by the model anyway.

"Relatively" here was about a comparison between nighttime and daytime BLHs. "Relatively" has been deleted

33. P10, line 27: costal should be coastal.

Corrected

34. P10, line 31: As mentioned in previous comments, figure 2, and, in general, ceilometer related data, should be removed in this paper (the correlation is poor and subject to many discussions that are not appropriate to discuss in this type of paper). Furthermore, this figure

that shows observations vs. observations does not fit in a section which is called simulations vs. observations?

As discussed in our reply to the reviewer's major comments (see item 5), we prefer to keep the analysis based on the ceilometer/lidar data, but moving the Figure 2 (submitted version of the paper) in the Supplement (Figure S1 in the revised version). The text has been revised accordingly

35. P10, line 37: DeBilt should be De Bilt.

Corrected

36. P10, Figures 6 and 7 are redundant, see one of the major comments.

We prefer to keep these two Figures, but they are now in a single Figure (Figure 5 in the revised version). For more detail, see our reply to the reviewer's major comments (item 5)

37. p10, lines 29 to 45 should be removed. See previous comments on ceilometer data.

In accordance with our reply to the reviewer's major comments (see item 5) about the use of ceilometer/lidar in this paper, we have kept the content of this part of the text

38. P11, line 4-5: About the timing of Rn-concentrations (05 and 14 UTC). Does this hold for both summer and winter?

The chosen reference times should be reasonable for both summer and winter seasons

39. P11, line 14: The list of coastal stations gets longer throughout the paper. Add it as labels in the paper. Is Cabauw really coastal station? I don't think locals would agree.

See our reply to the reviewer's major comment (6)

40. P11, line 24: This could be the start of an additional section.

Yes. We add a new sub-section: "Relationship between 222Rn activity concentrations and BLHs"

41. P11, line 30: What do you mean by "Apparently"?

The term "apparently" has been deleted

42. P11, line 31: What's a "model world"?

The sentence has been changed as follows: "The sharp changes in BLHs and 222Rn activity concentrations are due to the relatively coarse temporal resolution of ECMWF meteorological data (3-hourly for surface data (e.g., BLHs) and 6-hourly for 3D fields (temperature, wind, humidity, and convection); see Section 3.1)"

43. P11, line 44: "This finding..." What finding exactly?

This "finding" refers to the fact during daytime, the TM5 BLHs are close to IGRA measurements at most stations, while larger differences are observed between 222Rn activity concentration simulated and observed. The text has been clarified

44. P9-13, Section 5. what about a station selection for the figures? You seem to overwhelm the reader with graphs and bars, whereas only few things are to be highlighted. For example, the coastal and non-coastal zones are interesting, some are necessary to show due to later analysis with the Rn- and BLH combination. But certainly not all the stations are necessary. Then space would be saved, figures could be enlarged, these would be better readable and the article in general would be better appreciated. All the other redundant stations can then be stored in the supplemental material (which is very large as well).

See our reply to the reviewer's major comments (item 7). We have considered in the revised version of the paper only few relevant experimental settings. This makes the figures more readable

**45. P10-11, Section 5: Can be better divided in more sections.**

Yes. We add four more sub-sections as follows:

- Relationship between 222Rn activity concentrations and BLHs
- Sensitivity of simulated 222Rn activity concentrations to convection scheme
- Comparison of simulated and observed 222Rn activity concentrations: Impact of sampling time
- Vertical gradients of 222Rn activity concentrations in the boundary layer at Cabauw

46. P12, line 11: Add section 5.4 for this paragraph.

Yes. See our responses above

47. P12, line 38: "weather conditions", I suppose you mean "stability regimes"?

Yes. Corrected and suggested

48. P12, line 42-43: About CB1 and CB4, see remarks about table.

The text has been revised. The Table 1 is quoted there, where CB1 and CB4 are defined

49. P13, line 14 (section 6): this section feels more like a summary than a conclusion.

This has been revised

50. P13, line 16: "dynamics" should rather be "height". The dynamics are not evaluated.

See our reply to the reviewer's major comments (item 2). Therefore, we prefer to keep the term "dynamics"

51. P13, line 19: 10-20%, this is the first time I see a statistical value between TM5 and observations. That's very late.

Figure 11 (submitted version; Figure 8 in the revised version) that summarizes the statistics on BLHs and 222Rn activity concentrations is now commented earlier (see Section 4.2)

52. P13, line 23: IGRA observations (not "data").

Changed as suggested

53. P13, line 26: moderate correlation or reasonable? In any case, it is not good and opens up the floor for many discussions that are not relevant to the topic of the paper. See comments before about removing ceilometer related analysis.

We have kept the ceilometer/lidar observations in the analysis, but this part has slightly been revised

54. P13, lines 26-35: remove ceilometer related analysis/results

See our reply to the reviewer's major comments (see item 5)

55. P14, line 23-24: It is indeed difficult to draw conclusions. Try to be more quantitative, perhaps add more statistics, and this would make it easier to draw conclusions.

As clearly shown in Figure 11 (submitted version; Figure 9 in the revised version), the performance of the model simulations compared to the 222Rn activity concentration observations is very similar in term of e.g., root mean square and correlation coefficient for both convection schemes. The statistics are shown in Figure 11 (submitted version; Figure 8 in the revised version)

56. Table 1. Extend this table with labels as coastal and non-coastal stations. what is ANSTO? What is CB1? CB4? Probably different levels at the tower? The average readers do not have previous knowledge of the dataset and an attempt needs to be made to make it more readable for them. Maybe you want to change CB1 and CB4 to level 1 and level 2. Are the 'o' for latitude and longitude degree (o) signs? What does Altitude/Height exactly mean? Do you mean Terrain elevation (Mean Sea Level) and Height (Above Ground Level), respectively?

ANSTO stands for Australian Nuclear Science and Technology Organisation and was already defined in the submitted version of text (Section 2.2). The Table 1 (including its caption) has been revised (including definition of ANSTO) to clarify the points mentioned in the comments. However, as already mentioned, we do not qualify the stations as "coastal or non-coastal sites"

57. Figure 1. What are the abbrevations? The authors should refer to Table 1. Some letters are very hard to read, consider another color for either the names, or for the black dots. The black triangles and orange circles are barely visible. Where are the coastal stations exactly? What are the vertical and horizontal lines? Longitude and latitude? It should be indicated on the axis.

Figure 1 (including the caption) has been revised. Table 1 is now referred

58. Figure 2. remove. See major/minor comments

Figure 2 (submitted version) is now put in the Supplement (Figure S1 in the revised version)

59. Figure 3. Vertical and horizontal axis in the upper diagram?

They are latitude and longitude.

60. Figure 4. These figures are very small. Can the whisker plots be centralized around the months to which they are concerning to? And maybe the spacing then between the whisker plots could be enlarged. The scaling on y-axis is not the same. It doesn't have to, but it should be stated in the caption. Although the scales are not very far apart, so probably same axis length would work. This is true for almost all figures.

Figures 4-5 (submitted version; Figures 3-4 in the submitted version) have been revised. The scaling on y-axis is now the same for nocturnal BLHs and different, but the same for daytime BLHs. In general most of the figures of the paper have been revised, as mentioned above

61. In the text it is referred to coastal and non-coastal stations. It would be helpful to highlight that in the figure. Is it really necessary to show all stations? Maybe you could make coastal stations fig4a and continental stations in fig. 4b?

Figure 1 in the revised version helps to have an idea about coastal and non-coastal stations, as already stressed on our reply to the reviewer's major comments (see item 6)

62. Figure 6/7. Redundant/remove

See our reply to the reviewer's major comments (5)

63. Figure 11: remove CEIL/LIDAR data from figure.

See our reply to the reviewer's major comments (item 5).

64. Figure 12. What is ratio? TM5 divided by IGRA?

Yes. This has been clarified. It is now Figure 10 in the revised version of the paper

65. Figure 13. Abbreviations are not explained well in caption (e.g what is CB1 and CB4?).

"Mean diurnal variations..." should probably be "Monthly mean diurnal variations..."

Corrected by "Monthly mean diurnal variations" as suggested.

The acronyms CB1 and CB2 already defined in Table 1. These acronyms are now defined in the caption of Figure 13 (submitted version; Figure 11 in the revised version)

66. Figure 13: Data points are outside the y-axis range. This should be corrected.

Yes, but we did so because we wanted to focus on the variations of the vertical gradients of 222Rn activity concentrations around noon. We now increase a bit the y-axis, but some data points during night and early in the morning are still outside

67. Figure 14. How can the R of the lower figure be almost the same as the R for the upper figure?

This has been verified and it is correct. This is fortuitous. This is certainly due to the large values for daytime

**26 4.1. Model boundary layer heights**

| We extract the TM5 BLHs using either the TM5 default expression of R ib (Section 3.2),             |
|---------------------------------------------------------------------------------------------------------------|
| representing the effective BLH in the TM5 simulations, or based on Seidel et al. (2012) used in               |
| the InGOS model validation exercise (i.e., R ie = 0.25 and both surface wind and friction velocity |
| are set to zero in Eq.1; see Section 3.2). Furthermore, because InGOS and IGRA sites are not co-              |
| located, we extract the BLH in the model both at the location of the InGOS station and at the                 |
| location of the nearest IGRA station, resulting in a set of four different modeled BLHs labelled              |
| by the following acronyms:                                                                                    |
| • 'TM5': TM5 default version (Eq.1 in Section 3.2 with Ric =0.3); extracted at InGOS                          |
| stations by using 2D interpolation                                                                            |
|                                                                                                               |

- 'TM5\_IGRA': As 'TM5', but extracted at IGRA station, which is closest to the selected InGOS station
- 39 'TM5\_INGOS': BLHs computed in TM5 model adopting the InGOS definition of the 40 BLH (i.e., Rie = 0.25 and both surface wind and stress velocity are set to zero in Eq.1), 41 extracted at InGOS station.
  - 'TM5\_INGOS\_IGRA': As 'TM5\_INGOS', but extracted at IGRA station, which is closest to the selected InGOS station

45 Furthermore, we evaluate the BLHs as provided by ECMWF analyses and interpolated to TM5
 46 grids (labelled 'ECMWF'). The values of these BLHs are extracted only at the InGOS stations.

 The ECMWF BLH is determined using an entraining parcel method, selecting the top of stratocumulus, or cloud base in shallow convection situations (Dee et al., 2011).

4 5 | 4.2. 6 7 We 8 | soil

3

12

13

14

15

16

17

18

19

20 21

22

23 24

25

26 27

31

4.2. 3.4. Simulated 222Rn activity concentrations

[revised manuscript text omitted]

 222Rn concentrations at the 10 studied InGOS stations (with 222Rn activity concentration observations available) for 2009.

**26 Relationship between 222Rn activity concentrations and boundary layer heights**

27

25

1 2

In the following, we analyze the relationship between 222Rn activity concentration and BLH in 28 more detail. Figure 910 shows the mean seasonal diurnal cycle of observed and simulated 222Rn 29 activity concentration and BLH for the four seasons at different sites. The figure illustrates the 30 very strong anti-correlation between simulated BLH and 222Rn activity concentration: The 31 modelled BLHs increase sharply between 9:00 and 10:00 UTC (10:00/11:00 and 11:00/12:00 32 LT), resulting in an immediate decrease of modelled 222Rn concentrations. In contrast, the 222Rn 33 activity concentration measurements show a slower decrease over several hours. Although this 34 slow decrease may be partially due to the slow (45min) response time of the two-filter detectors, 35 it is clear that the sharp changes in simulated BLHs and 222Rn activity concentrations are due to 36 the relatively coarse temporal resolution of ECMWF meteorological data (3-hourly for surface 37 data (e.g., BLHs) and 6-hourly for 3D fields (temperature, wind and humidity); see Section 38 3.1). Apparently the sharp changes in the 'model world' are due to the relatively coarse temporal 39 resolution of ECMWF meteorological data (3-hourly for surface data (e.g., BLHs) and 6-hourly 40 for 3D fields (temperature, wind and humidity); see Section 3.1). Because the ceilometer data at 41 Cabauw during night might be questionable, we included in Figure 910 only the lidar 42 measurements at Trainou (TR4). These-that shows a much slower growth of the BLH, starting in 43 the morning and reaching its maximum in the late afternoon, as also illustrated in Pal et al. 44 45 (2012, 2015). DespiteIn spite of the obvious issue of the temporal resolution of the model, however, inspection of **Figure 910** also indicates significant illustrates that the mismatches 46

between simulated and observed 222Rn activity concentrations that cannot be explained wholly 1 by problems with the modeled BLH (even accounting for possible instrumental response time 2 effects). Especially during daytime, the TM5 BLHs are close to the IGRA measurements at most 3 4 stations (as also illustrated by the ratios of BLHs in Figure 8), whereas while larger differences are observed between the simulated and measured 222Rn activity concentrations simulations and 5 measurements at several stations. This is further illustrated in Figure 1011, where we compare 6 7 the ratio of simulated toand observed BLH with the ratio of observed to simulated and observed 222Rn activity concentration during daytime, and in Figure 12 where these ratios are shown for 8 the different seasons. If the 222Rn activity concentration errors were purely due incorrect 9 dilutions resulting from errors in the modeled BLH at a given station, the two ratios would be 10 similar. This is clearly not the case, however, and the modelled afternoon concentration ratios 11 range widely (from 0.2 to 1.8) from station to station. These This mismatches between observed 12 and simulated 222Rn activity concentrations may be related to finding points to potential 13 shortcomings of TM5 into correctly simulatinge the vertical 222Rn 
[revised manuscript text omitted]

| (CB1)      |            |             |                   |                   |                     |                   |                                  |
| CBW        | Cabauw     | Netherlands | 51.97             | 4.93              | 199/200             | ANSTO             | Vermeulen et al. (2011)          |
| (CB4)      |            |             |                   |                   |                     |                   |                                  |
| EGH        | Egham      | United      | 51.43             | -0.56             | 45 70/10     | one filter method | Levin et al. (2002)              |
|            |            | Kingdom     |                   |                   |                     |                   |                                  |
| GIF        | Gif-sur-   | France      | 48.71             | 2.15              | 167/7               | one-filter method | Lopez et al. (2012), Yver et al. |
|            | Yvette     |             |                   |                   |                     |                   | (2009)                           |
| HEI        | Heidelberg | Germany     | 49.42             | 8.71              | 146/30              | one-filter method | Levin et al. (2002)              |
| TRN (TR4)  | Trainou    | France      | 47.95             | 2.11              | 311/180             | ANSTO             | Schmidt et al. (2014)            |
| IPR        | Ispra      | Italy       | 45.80             | 8.63              | $223/3.5(15)^{1}$   | ANSTO             | Scheeren and Bergamaschi         |
|            |            |             |                   |                   |                     |                   | (2012)                           |

1measurements at 3.5m 'normalized' to sampling height of 15m based on wind-speed dependent correction (see Section 2.2)